# CombFold: predicting structures of large protein assemblies using a combinatorial assembly algorithm and AlphaFold2

**Ben Shor** & **Dina Schneidman-Duhovny** ✉

Deep learning models, such as AlphaFold2 and RosettaFold, enable high-accuracy protein structure prediction. However, large protein complexes are still challenging to predict due to their size and the complexity of interactions between multiple subunits. Here we present CombFold, a combinatorial and hierarchical assembly algorithm for predicting structures of large protein complexes utilizing pairwise interactions between subunits predicted by AlphaFold2. CombFold accurately predicted (TM-score >0.7) 72% of the complexes among the top-10 predictions in two datasets of 60 large, asymmetric assemblies. Moreover, the structural coverage of predicted complexes was 20% higher compared to corresponding Protein Data Bank entries. We applied the method on complexes from Complex Portal with known stoichiometry but without known structure and obtained high-confidence predictions. CombFold supports the integration of distance restraints based on crosslinking mass spectrometry and fast enumeration of possible complex stoichiometries. CombFold's high accuracy makes it a promising tool for expanding structural coverage beyond monomeric proteins.

Most proteins function as multimolecular assemblies in the cells. There are on average a few dozen interactions per protein[1–3]. These assemblies perform important functions, such as energy transduction[4], transport[5] and signal transduction[6]. The determination of the 3D structures of these assemblies is critical for understanding their function and evolution, interpreting the effects of mutations, and potential applications in drug discovery. The large size of some assemblies and conformational heterogeneity pose challenges for traditional structural characterization techniques, such as X-ray crystallography and nuclear magnetic resonance spectroscopy. While progress has been made using cryo-electron microscopy (cryo-EM), high-throughput structure determination of large assemblies is still challenging.

Recently deep learning techniques greatly advanced our ability to predict high-accuracy protein structures. One of the most notable advancements was the release of AlphaFold2 (ref. 7) and RosettaFold[8]. While AlphaFold2 was designed to predict single-chain proteins, it can also apply to predict protein complexes using the same architecture. Soon after its release, several techniques were developed to use AlphaFold2 to predict multichain protein complexes—first by using a linker[9] and later by offsetting the residue index[10]. Similar techniques were used for the training of AlphaFold-Multimer (AFM)[11] which is able to predict multimeric complexes with high accuracy using paired and padded multiple sequence alignment. On several pairwise protein–protein docking benchmarks AFM achieves a success rate of 40–70% for complexes consisting of two to nine chains up to 1,536 in total length[11–13].

However, AFM application for predicting structures of large assemblies is still challenging[12,13]. The first difficulty is the requirement for substantial resources, such as a graphical processing unit (GPU) with a large memory size. Currently, common GPUs have ~20 GB of memory, enabling the prediction of complexes up to 1,800 and 3,000 amino acids for AFM version 2.2 (AFMv2) and AFM version 2.3

The Rachel and Selim Benin School of Computer Science and Engineering, The Hebrew University of Jerusalem, Jerusalem, Israel.
✉e-mail: dina.schneidman@mail.huji.ac.il

(AFMv3), respectively. We estimate that in a few years GPU cards with sufficient memory will become widely available. However, as AFM memory usage increases roughly quadratically with the number of amino acids[7], this currently limits the practical capability of many researchers to predict structures of large size, leaving many macromolecular complexes without a structure prediction. The second difficulty is sampling with a large number of restraints: as the number of chains and amino acids increases, the number of residue–residue contacts and distance restraints to optimize increases as well, making it harder for the model to converge to accurate structures. Large, multimolecular complex prediction is an out-of-domain inference setup for AFM since it was trained only on cropped regions and thus is not expected to perform well. The third difficulty is that AFM converges to a single (sometimes incorrect) structure (for each of the five available trained models) and it is highly challenging to obtain a diverse set of predictions for the same target[14].

Prior to the deep learning revolution, methods developed for the assembly of multiprotein complexes could be divided into two main categories. The first category is integrative modeling methods that mainly rely on experimental data[15,16], and the second is docking-based methods that rely on pairwise protein–protein docking[17–19]. Integrative modeling methods rely on information from multiple sources, such as crosslinking mass spectrometry, Förster resonance energy transfer (FRET), co-evolution, cryo-EM and small-angle X-ray scattering to compute models. This information is converted into spatial restraints and combined into an integrative modeling approach[20,21], using specialized software packages[22–24] to generate a set of structural models that are consistent with it. The integrative modeling workflow iterates through four stages that convert input information into an output model: (1) gathering data, (2) scoring (representing and translating the data into spatial restraints), (3) sampling, and (4) validating the model[15,22]. The sampling of candidate models is often performed by global data-driven optimization algorithms, such as Monte Carlo or genetic algorithms. The input information contributes to a scoring function, either for ranking or filtering generated structural models or for directly guiding the sampling process. Integrative structure modeling is applicable to large and heterogeneous systems[25], such as the ~52 MDa nuclear pore complex[26]. AlphaLink[27] was developed recently to support such sampling with distance restraints using AlphaFold2.

The second category of docking-based methods predominantly rely on pairwise protein–protein docking for the prediction of complexes[28–31] and do not require additional input information. In pairwise docking, the two input proteins are docked to one another using geometric shape and physicochemical complementarity. The main problem is that they sample thousands of docked configurations. While the correct ones are usually sampled, it is difficult to rank them as top-scoring. Typically, pairwise docking methods succeed in ranking a correct model among the top-10 best scoring in 25–40% of the cases[32,33]. This low accuracy further complicates the multiprotein assembly stage, where methods have to consider a large number of pairwise protein–protein docking models. For example, Multi-LZerD[18] builds the multimolecular assembly by applying a stochastic search driven by a genetic algorithm. Kuzu et al.[19] construct the multimolecular complex iteratively, where a single subunit is added to the subassembly in each iteration. The CombDock method is hierarchical and combinatorial[17,34]. The complexes are constructed hierarchically by generating subassemblies of two or more subunits. At each stage, subassemblies are connected using pairwise docking configurations between subunits. Due to multiple possible hierarchical assembly pathways, the algorithm combinatorially enumerates assembly trees. Since the algorithms used for docking and scoring pairwise interactions have low accuracy, it is difficult to reach high accuracy in multisubunit docking.

The recently developed MoLPC method relies on AlphaFold2 to produce configurations for pairs and triplets of chains and assemble them using Monte Carlo Tree Search[35]. However, the approach is applicable mainly to homomeric complexes with a success rate of ~30%. In this Article, inspired by this work, we combine AlphaFold2 with a deterministic combinatorial assembly algorithm[17,34]. Our new method, CombFold, uses a small number of pairwise subunit interactions generated by AlphaFold2 for assembly instead of thousands generated by docking. The hierarchical and combinatorial assembly stage exhaustively enumerates possible assembly trees, maximizing the probability of correctly assembling the complex based on pairwise AlphaFold2 interactions. We validate our approach on two benchmarks of large heteromeric assemblies (up to 30 chains and 18,000 amino acids) and obtain a top-1 success of 62% and top-10 success rate of 72% (TM-score >0.7). Moreover, CombFold is able to increase the structural coverage by 20% relative to experimental structures in our benchmarks. Integration of distance restraints based on crosslinking mass spectrometry further increases the success rate. We also test the method on the benchmark of homomeric complexes used for MoLPC validation and obtain a top-1 success rate of 57%. CombFold successfully assembles six out of seven CASP15 targets with over 3,000 amino acids (Supplementary Note 1 and Supplementary Fig. 1). We apply the method on a set of complexes with known stoichiometry and without known structure from Complex Portal[36] and obtain confident predictions.

## Results

### Overview of CombFold

The input to CombFold is the subunit sequences and optionally distance restraints, the output is a set of assembled structures. A subunit can be a single chain or a domain. The approach is based on combinatorial and hierarchical assembly via pairwise interactions. In principle, there is no limitation on complex size, as the complex can be divided into subunits suited for the GPU memory limit, and our current implementation supports up to 128 subunits. CombFold works in three major stages: (1) generation of pairwise subunit interactions by AFM, (2) creation of a unified representation of subunits and interactions, and (3) combinatorial assembly of subunits (Fig. 1).

In the first stage, we apply AFM to all possible subunit pairings. Following this, we create three additional AFM models for each subunit, ranging in size from three to five subunits, that include subunits with which the given subunit had the highest confidence-scored predicted pairwise interactions (Methods). The underlying concept is that some groups of more than two subunits form intertwined structures, and therefore all of them should be predicted as a single model by AFM (Methods).

In the second stage, to prepare input for the third assembly stage, a single representative structure for each subunit is selected and the transformations between representative subunits are calculated. This is required since there are multiple AFM structures for each subunit from pairwise AFM runs and their enumeration during the assembly stage is intractable. The representative subunit structures are extracted from the predicted modeled subcomplexes according to the maximal average predicted local Distance Difference Test (plDDT) score for this subunit. Next, we use all interacting subunit pairs (Cα–Cα distance <8 Å) from AFM models to extract pairwise transformations (rotation and translation in 3D) between their representative structures in the global reference frame. The representation of the input by representative subunit structures and transformations between them enables us to apply the combinatorial assembly algorithm with AFM interactions instead of docking-based ones. Each transformation is coupled with a score based on AFM's predicted aligned error (PAE) score (Methods).

In the third stage, we use N representative subunit structures, the pairwise transformations between them and, optionally, distance restraints for the hierarchical and combinatorial assembly of the entire complex. Distance restraints can originate from crosslinking mass spectrometry, FRET or other sources of information[37–40]. If a protein chain is divided into subunits (for example, domains), distance constraints are

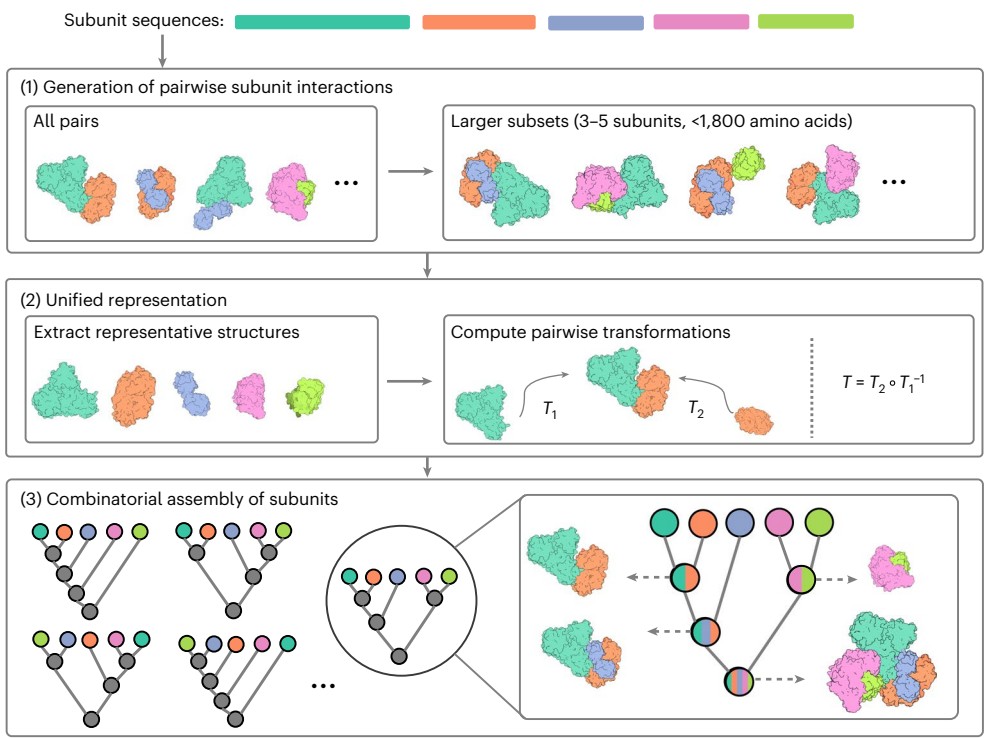

**Fig. 1 | The three stages of the CombFold assembly algorithm.** The input is the sequences of the subunits in the complex. (1) Structure prediction of all pairwise and some larger subunit subsets using AFM. (2) Selection of representative subunit structures out of all predicted structures, followed by computation of all pairwise transformations present in predicted structures relative to the representative structures. (3) Combinatorial and hierarchical assembly of subunit structures using the computed pairwise transformations. In each iteration, new subcomplexes are assembled using a pairwise transformation to join two previously created subcomplexes.

added to enforce sequence connectivity. This combinatorial assembly stage consists of $N$ iterations, where in the $i$th iteration we construct $K$ subcomplexes of size $i$. The value of $K$ has to be large enough to contain a variety of subcomplexes. Subcomplexes of size $i$ are constructed from pairs of previously computed subcomplexes of size 1 to $i − 1$. For example, a subcomplex of size $i$ can be computed by merging subcomplexes of size 3 and $i − 3$. We attempt to merge a pair of subcomplexes if they do not have any shared subunit and the joint number of subunits is $i$. During the merge, new subcomplexes are generated by iterating all subunit pairs (one from each subcomplex) and applying known transformations between those two subunits on the entire subcomplexes. Next, we discard generated subcomplexes with major steric clashes or chain connectivity violations. Distance restraints satisfaction is calculated, and low-scoring subcomplexes are also discarded (Extended Data Fig. 1). The remaining subcomplexes are clustered and scored on the basis of the score of transformations that were used, and the top $K$ subcomplexes are saved for the next iterations.

The model confidence score produced by our method is based on the AFM PAE score. Each pairwise interaction (represented by a transformation) has a PAE-based score (Methods). The confidence of an assembled structure is a weighted score of the transformations that were used for assembly, where the weight is proportional to the sizes of the subunit subsets that were merged by each transformation.

**Benchmark datasets.** We tested the method on four benchmark datasets (Table 1 and Supplementary Note 2). We generated a Benchmark 1 dataset aimed to test the method on large heteromeric complexes. Structures with many unique chains usually do not contain notable symmetry which makes them more challenging for assembly, since many different pairwise interactions need to be found and combined. Benchmark 1 contains 35 structures with 5 to 20 chains and at least 5 unique chains per complex, consisting of 1,300 to 8,000 amino acids

(Extended Data Fig. 2a). This dataset includes only complexes released after April 2018, which AFMv2 was not trained on. Benchmark 2 dataset was generated similarly to Benchmark 1 to test the recently released AFMv3. It contains 25 complexes with 5–30 chains and 2,000–18,000 amino acids (Extended Data Fig. 2b) that were not in the training set of AFMv3 (released after September 2021). Benchmark 3 dataset was used for benchmarking the MoLPC approach[35]. It contains 153 complexes ranging between 500 and 10,000 amino acids with 10–30 chains per complex. This dataset contains mainly symmetric homomers (98 complexes consisting of one unique chain and 27 consisting of two unique chains). Finally, Benchmark 4 dataset contains seven CASP15 targets with more than 3,000 amino acids.

**Accuracy assessment.** To evaluate the accuracy of the modeled structures we rely on the TM-score[41] which assesses the global accuracy of the complex, similar to CASP and MoLPC[35]. Similarly to CAPRI assessment[42], a model is considered acceptable quality if the TM-score is above 0.7 and high quality if the TM-score is above 0.8. The success rate is measured as a fraction of the benchmark complexes with acceptable- or high-quality models among the top-$N$ best-scoring predictions.

**Accuracy on Benchmark 1 (heteromers).** We obtain a top-1 success rate of 60% for CombFold on this benchmark, accurately modeling 21 out of 35 complexes (Fig. 2a) with TM-score >0.7. High-quality top-1 models are produced for 14 complexes (40%). When considering the top-10 models, the success rate is 74%. Importantly, the predicted confidence correlates with the TM-score (Pearson $r = 0.57$, Fig. 2b), indicating that it can be used to estimate model accuracy. To determine to which extent the success rate depends on the ability of AFM to produce accurate models for pairwise interactions, we calculate the pairwise connectivity (Methods). As expected, the pairwise connectivity correlates with the TM-score (Pearson $r = 0.48$, Fig. 2c).

**Table 1 | CombFold evaluation benchmarks**

| Benchmark | Complex type | Number of complexes | Number of chains | Number of amino acids | Top-1 success rate of CombFold | Top-10 success rate of CombFold | Top-1 success rate of AFM or MoLPC |
|---|---|---|---|---|---|---|---|
| 1 | Asymmetric complexes (released after AFMv2 training) | 35 | 5–20 | 1,300–8,000 | 60% | 74% | 26% (AFMv2) |
| 2 | Asymmetric complexes (released after AFMv3 training) | 25 | 5–30 | 2,000–18,000 | 64% | 68% | 36% (AFMv3) |
| 3 | Mostly homomers and symmetric complexes | 153 | 10–30 | 600–10,000 | 57% | 58% | 28% (MoLPC) |
| 4 | CASP15 targets (>3,000 amino acids) | 7 | 1–27 | 3,000–8,000 | 57–86%[a] | 57–86% | 43% (AFM, MoLPC[b]) |

The success rate is defined as the fraction of benchmark cases with a model with a TM-score above 0.7 among the top-*N* best-scoring predictions. [a]For CASP15 targets the fully automated CombFold had a success rate of 57%. Manual subdivision of proteins into domains led to an increased success rate of 86%. [b]We compared CombFold to CASP15 submissions of the Elofsson group that used AFM and MoLPC.

We compare CombFold to an end-to-end AFM on all the Benchmark 1 complexes using the A100 GPU card with 40-GB memory. AFM succeeded in producing at least one result for 17 out of 35 complexes with up to 3,700 amino acids. Of these, ten complexes were modeled with acceptable or high quality, resulting in success rates of 26% and 29% for top-1 and top-5 results, respectively (Fig. 2a,d).

The largest complex assembled by CombFold was eIF2B:eIF2 (Protein Data Bank (PDB) 6I3M, Fig. 2e), which could not be assembled directly with AFM. The CombFold model contains a structural coverage for 6,114 amino acids with plDDT above 50 out of a total of 7,486. In comparison, the experimental cryo-EM structure covers only 4,680 amino acids. The addition of over 1,500 amino acids contains six well-folded domains. This example demonstrates the ability of CombFold to complete unresolved fragments in experimental structures. On average, each assembled complex in this Benchmark contained 20% more amino acids compared to the corresponding PDB entry. GID E3 ubiquitin ligase complex is another example where an additional domain is missing in the experimental structure (PDB 6SWY, Fig. 2f) and is predicted by CombFold with high plDDT. The complex is assembled with a TM-score of 0.83 compared to AFM, which produces a model with a TM-score of 0.53. In contrast, the multiple resistance and pH adaptation (Mrp) complex (PDB 7D3U, Fig. 2g) is assembled with higher accuracy by AFM (TM-score 0.97 versus 0.67 for CombFold). This is due to the fact that the orientation between the two domains in the largest subunit was not accurately predicted in the representative structure chosen for assembly (Fig. 2g, light blue).

**Accuracy on Benchmark 2 (heteromers).** This benchmark was generated to test CombFold against the recently released AFMv3. We also used AFMv3 to predict the pairwise subunit interactions for CombFold (instead of AFMv2 in Benchmark 1). The performance on this dataset is comparable to Benchmark 1 (Extended Data Fig. 3), with top-1 and top-5 success rates of 64% and 68%, respectively. In comparison, the top-1 success rate of AFMv3 is 36%. The fraction of high-quality top-1 models is higher on this Benchmark (52% versus 40% for Benchmark 1), indicating that AFMv3 produces pairwise interactions with higher accuracy (Extended Data Fig. 4), perhaps due to the higher number of recycles and larger training set. To further validate CombFold, we used this Benchmark for comparison to RosettaFold2 (ref. 43). RosettaFold2 was not able to assemble most complexes (21/25), and among the assembled four complexes, only one had an acceptable-quality model among the ten predicted structures, which translates to a success rate of 4% (Extended Data Fig. 3a).

While TM-score is a measure of global accuracy, to assess the accuracy of subunit interfaces, we calculate the interface contact similarity (ICS) score[44] that is also used in the CASP/CAPRI complex assessment. Similarly to the TM-score, ICS values are in the range of 0–1; however, the ICS scores are usually lower compared to TM-scores, indicating that a model with high global accuracy may still have low-quality interfaces

and contacts. We find that CombFold top-1 models have variable ICS scores (Extended Data Fig. 3e). Moreover, AFM models have higher scores compared to CombFold. The lower ICS scores of CombFold can be attributed to the usage of representative subunit structures instead of the ones produced by pairwise AFM. In addition, some of the interfaces in the CombFold models are not a result of pairwise AFM prediction, but a by-product of the assembly process, and therefore have lower quality.

We examine whether the interface quality of CombFold models is sufficient for predicting dissociation constants ($k_D$) between subunits. Because experimentally measured $k_D$ values are not available for the whole Benchmark, we compare the $k_D$ values predicted by PRODIGY[45] from the interfaces in experimental structures to the $k_D$ values predicted from the interfaces in the top-1 model of CombFold. We find a strong correlation (Spearman $r = 0.55$, Extended Data Fig. 3f), indicating that despite lower ICS scores, CombFold models are sufficiently accurate for estimating $k_D$.

**Integration of experimental data.** Integrative structure modeling is often used to determine the structures of large macromolecular assemblies using information from a variety of sources, such as crosslinking mass spectrometry, cryo-EM or bioinformatics analysis[22,26,46–48]. The information is used for scoring and sampling models to produce structures that are consistent with the available data. Here we add to CombFold support for integrating information about known physical interactions between subunits and distance restraints that originate from crosslinking mass spectrometry. This type of information can be obtained for individual complexes in vitro or for multiple assemblies identified from in situ experiments[49–52]. AFM does not currently support the integration of this type of data. Recently, AlphaLink[27] was developed to add distance restraints support to AlphaFold2/OpenFold as a bias to residue–residue contacts, similar to template support in AlphaFold2. This method requires subsampling of multiple sequence alignment to give more weight to distance restraints and is currently applicable for complexes with less than 3,000 amino acids[53]. The advantage of CombFold is that it can integrate additional information during the assembly stage (Methods).

We apply CombFold with distance restraints for human mitochondrial translocase TIM22 (PDB 7CGP), a Benchmark 1 case, for which both CombFold and AFM failed to produce an accurate prediction (TM-score of 0.57 and 0.67, respectively). We used crosslinking mass spectrometry experiment for this complex[54] to compile a set of 12 distance restraints. We also divided the chains into two groups for assembly (Methods), based on a known structure of a subcomplex of TIM9 and TIM10 (PDB 2BSK). The resulting model is of high quality with a TM-score of 0.85 (Fig. 2h).

To further examine the contribution of crosslinking mass spectrometry data, we simulated crosslinks for Benchmark 2 (Methods) and compared the performance of CombFold with and without input

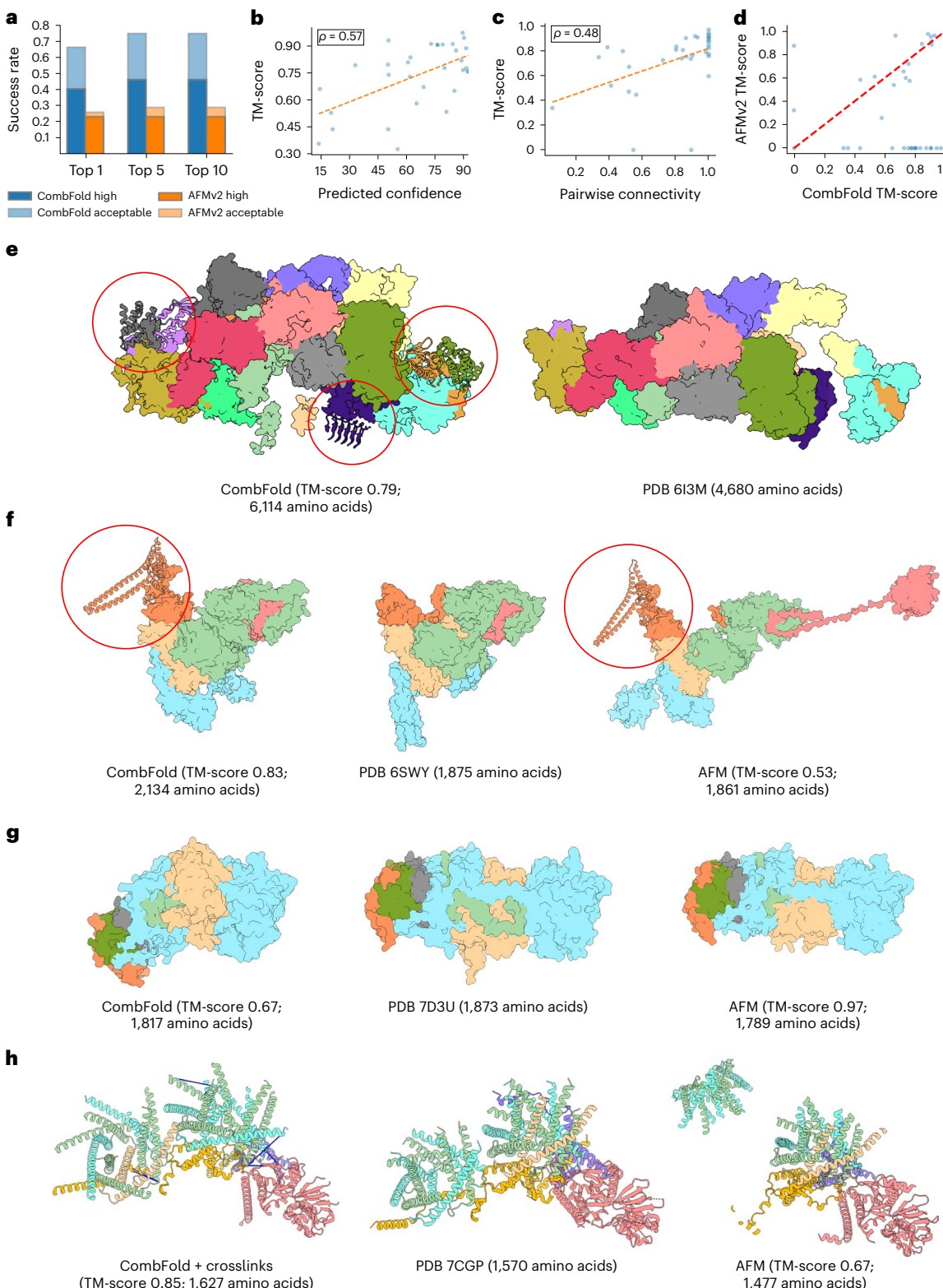

**Fig. 2 | Accuracy of CombFold on Benchmark 1. a**, The top-*N* (*N* = 1, 5, 10) success rate of CombFold (blue) and AFM (orange). AFM produces only five predictions. **b**, Predicted confidence versus the TM-score for CombFold. **c**, Success rate of AFM in producing pairwise interactions as measured by the pairwise connectivity versus the TM-score of the models produced by CombFold. **d**, TM-score of AFM models versus CombFold models. **e**, eIF2B:eIF2 complex: CombFold model (left) and cryo-EM structure (right). The model contains over 1,500 additional amino acids (marked with red circles). **f**, GID E3 ubiquitin ligase complex: high-quality CombFold model (left), cryo-EM structure (middle) and inaccurate AFM model (right). **g**, Multiple resistance and pH adaptation (Mrp) complex: inaccurate CombFold model (left), cryo-EM structure (middle) and high-quality AFM model (right). **h**, Human mitochondrial translocase TIM22: high-quality model by CombFold, integrating experimental crosslinking data (left), cryo-EM structure (middle) and inaccurate AFM model (right). Crosslinks are shown as blue lines.

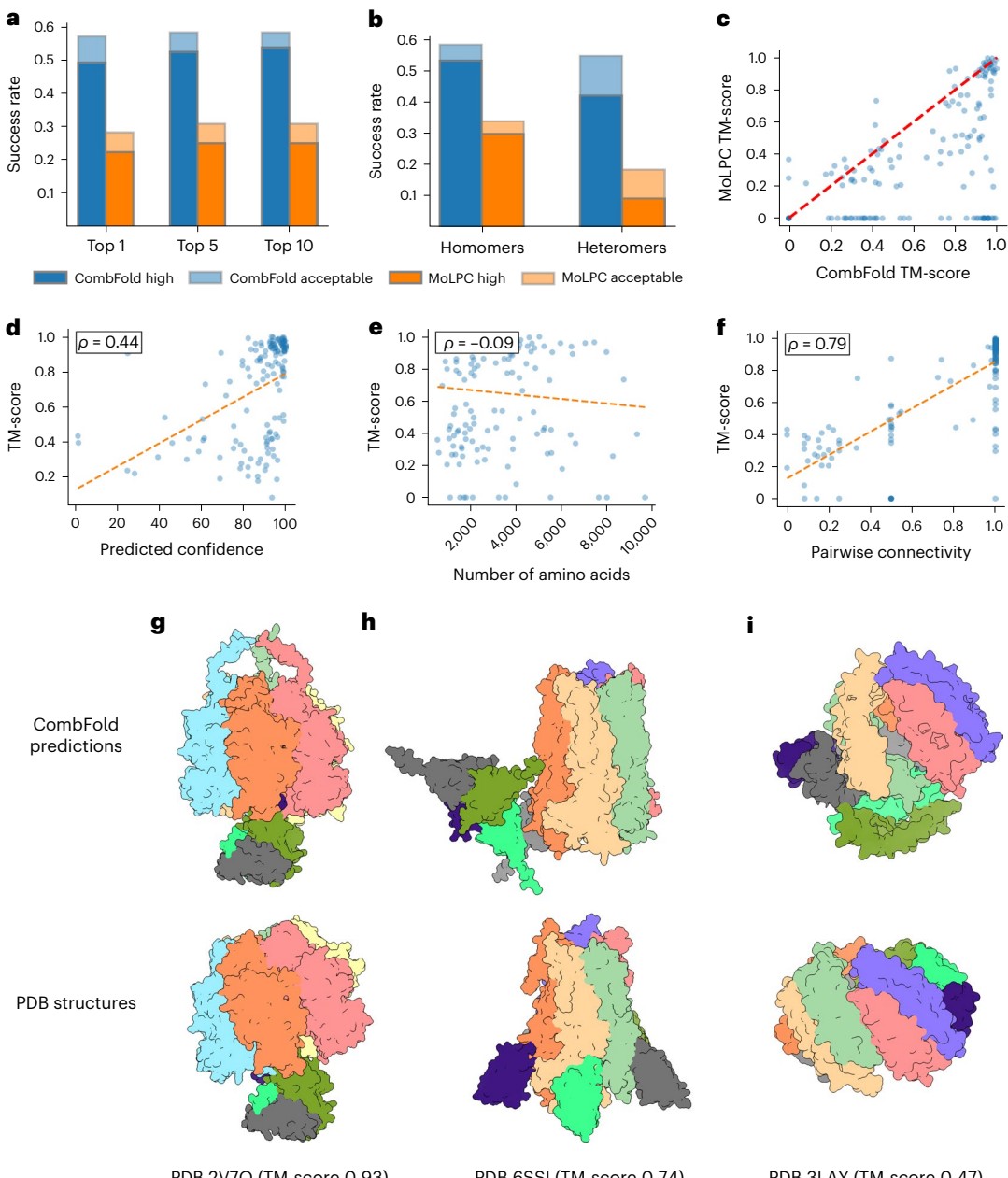

**Fig. 3 | Accuracy of CombFold on Benchmark 3. a**, The top-*N* (*N* = 1, 5, 10) success rate of CombFold (blue) and MoLPC (orange). **b**, Top-1 success rate for homomers and heteromers. **c**, TM-score comparison for CombFold and MoLPC. **d**, Predicted confidence versus the TM-score for CombFold. **e**, The number of complex amino acids versus the top-1 TM-score. **f**, The success rate of AFM in producing pairwise interactions as measured by the pairwise connectivity versus the TM-score. **g**, High-quality model of F1-ATPase (top) versus the X-ray structure (bottom). CombFold prediction contains 159 additional amino acids that are not modeled in the X-ray structure, providing full structural coverage. **h**, Acceptable-quality model of *Erwinia* ligand-gated ion channel in complex with nanobodies (top) versus X-ray structure (bottom). The channel is accurately modeled; however, the location of nanobodies is incorrect. **i**, Incorrect model of zinc resistance-associated protein from *Salmonella enterica* (top) versus X-ray structure (bottom).

crosslinks (Extended Data Fig. 3c,d). Integrating crosslinks increased the top-1 success rate to 76% (compared to 64% without crosslinks). We compared CombFold to AlphaLink[53] and HADDOCK[55] with the same set of crosslinks and obtained a success rate of 8% and 4%, respectively (Extended Data Fig. 3c,d).

**Accuracy on Benchmark 3 (mostly homomers).** We obtain a top-1 success rate of 57% on this benchmark, accurately modeling 87 out of 153 complexes (Fig. 3a and Table 1). Moreover, most of the successful predictions (75 out of 87) are of high quality (TM-score >0.8). When top-10 predictions are considered, the success rate is 58% and 82 out of 89 are of high quality. The higher fraction of complexes with

high-quality models compared to heteromeric Benchmarks 1 and 2 demonstrates the challenge of assembling heteromeric complexes with high accuracy where multiple intersubunit orientations need to be optimized simultaneously. The predicted confidence correlates with the TM-score (Pearson *r* = 0.44, Fig. 3d). Moreover, the accuracy of CombFold does not decrease with an increase in complex size (Pearson *r* = −0.09, Fig. 3e).

CombFold success rate correlates with the success of AFM in producing structures of pairwise interactions as measured by the pairwise connectivity (Pearson *r* = 0.79, Fig. 3f). This correlation is higher than for Benchmark 1 complexes, as in the assembly of homomeric structures, CombFold relies mainly on one or two pairwise interactions.

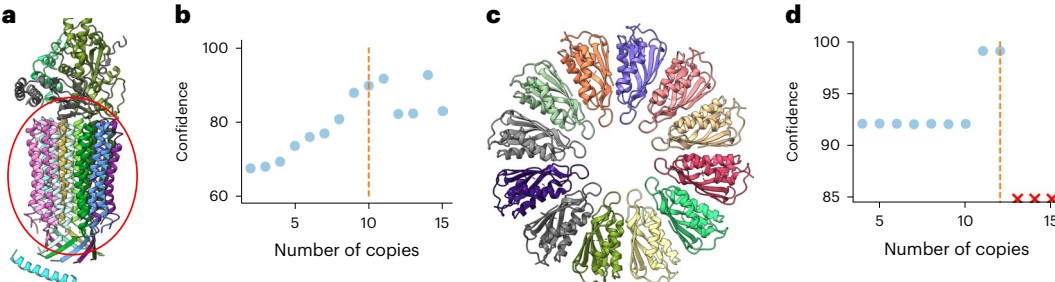

**Fig. 4 | Stoichiometry prediction. a**, A structure of mitochondrial ATP synthase with bound native cardiolipin (PDB 6TDX). Circled is a symmetrical structure formed from ten copies of subunit c. **b**, CombFold predicted confidence as a function of the number of copies of subunit c. **c**, A structure of PelC dodecamer (PDB 5T11). **d**, CombFold predicted confidence for PelC dodecamer as a function of the number of copies in input stoichiometry.

As a result, CombFold accuracy is limited by the reported success rate of ~60% for AFM in predicting pairwise protein–protein interactions[11–13]. In contrast, in the assembly of heteromeric structures, multiple pairwise interactions are considered, and pairwise interaction can form indirectly even if it is not predicted correctly by AFM. Therefore, the success rate of CombFold on heteromeric complexes is higher (Table 1 and Fig. 2). While heteromeric complexes are asymmetric by definition, they can include local symmetry resulting from multiple copies of one or more subunits[56]. Benchmark 3 contains four fully asymmetric complexes (without multiple subunit copies) and CombFold was able to assemble three with acceptable quality. The performance of CombFold on asymmetric structures is assessed on Benchmarks 1 and 2, which are almost entirely asymmetric (Extended Data Fig. 2).

For comparison, the top-1 success rate of MoLPC on Benchmark 3 is 28% and top-10 is 31% (Fig. 3a,c)[35]. This difference is attributed to our utilization of multiple AFM models and the assembly algorithm that performs a more exhaustive combinatorial and hierarchical search compared to the Monte Carlo Tree Search used by MoLPC. When Benchmark 3 complexes are divided into homomers and heteromers, there is no significant difference for our method, while there is a gap in favor of homomers for MoLPC (Fig. 3b).

**Application for predicting complexes without known structure.** Complex Portal is a database that contains manually curated information on stable macromolecular complexes[36]. We queried the database for all complexes with over 5,000 amino acids, known stoichiometry, and without homology to any experimentally determined structure (Methods) to obtain 28 complexes from three organisms (*Homo sapiens*, *Mus musculus* and *Saccharomyces cerevisiae*). High-confidence structures were found for seven complexes (Extended Data Figs. 5 and 6).

One of the high-confidence predictions is the human Elongator holoenzyme complex, which consists of six proteins, Elp1–6, two copies of each. A dimer of Elp123 subunits interacts with the Elp456 subcomplex. Partial homologous structures of *S. cerevisiae* are available, with larger subcomplexes published recently[57]. The structure predicted by CombFold is consistent with the published homologous structure (Extended Data Fig. 5a,b). Moreover, the predicted structure can be used to explain the effect of mutations. We extracted all the pathogenic mutations from ClinVar[58] (Supplementary Table 2) and classified them on the basis of the predicted structure into those that could disrupt protein core or protein–protein interactions (Extended Data Fig. 5c,d).

**Stoichiometry prediction.** The major obstacle to applying our method to known interactions and complexes is the need for stoichiometry information. Our assembly algorithm can be applied to a set of subunits without stoichiometry using the AFM-predicted representative structures and pairwise interactions as follows. Different stoichiometries can be enumerated using the same AFM models as an input and the confidence prediction can be used to estimate the correct stoichiometry.

This enables us to perform the resource-intensive AFM calculation once and sample possible stoichiometries with the fast assembly algorithm.

Here we present two examples of this application. The first is the complex of mitochondrial ATP synthase with bound native cardiolipin that contains ten copies of ATP synthase subunit c forming a symmetrical cylinder (Fig. 4a). We used CombFold to predict complexes with 14 stoichiometries: 2–15 copies of subunit c and the correct number of copies for all the other subunits. There is a significant increase in predicted confidence for assemblies with 10, 11 and 14 copies of subunit c (Fig. 4b), indicating that confidence can be used to narrow down the set of possible stoichiometries.

Another example is the PelC dodecamer from *Paraburkholderia phytofirmans*. This is a symmetrical complex composed of 12 copies of lipoprotein (Fig. 4c). We applied CombFold to predict complexes with 14 stoichiometries (2–15 copies of the PelC subunit). For 13 or more copies no structure could be assembled without major steric clashes. There is a spike in the predicted confidence for assemblies with 11 or 12 copies (Fig. 4d). This demonstrates not only that confidence is an indicator of stoichiometry, but that the ability to assemble is another indicator.

## Discussion

We present an approach to predict the structure of large multisubunit protein complexes based on substructures predicted by AFM for pairs or larger subsets of input subunits. Our method is powered by the combinatorial assembly algorithm that exhaustively enumerates best-scoring assembly trees resulting in accurately predicted assemblies. Moreover, information that can be converted into distance restraints, such as crosslinking mass spectrometry datasets, can be integrated into the assembly algorithm for higher accuracy (Extended Data Fig. 3c,d). We validate the approach on four datasets with top-10 success rate of 57–74% for both homomeric and heteromeric assemblies (Figs. 2 and 3, Table 1, Extended Data Fig. 3 and Supplementary Note 1). Moreover, CombFold is able to extend by 20% the structural coverage of experimentally solved large complexes where the modeled structure often does not fully cover the sequences. This enables the application of CombFold to extend the coverage of solved structures.

Most complexes could be assembled by CombFold using single chains as subunits. However, for some complexes, dividing chains into domain-level subunits is beneficial for correct assembly, such as CASP15 targets H1137 and T1169. While our method supports domain-level assembly, the decision of whether to split into domains is left to the user. Subcomplexes are often known based on prior knowledge or can be inferred from single-chain structures, such as intertwined domains in CASP targets H1137 and H1114. In these cases, our method can enforce the specific assembly order to compute the known subcomplexes followed by the generation of the whole assembly.

Currently, our success rate is limited by the ability of AFM to produce pairwise subunit interactions (Figs. 2c and 3f). In this regard, approaches that enhance the AFM sampling by enabling dropout at inference can be useful[14,59]. Additional pairwise orientations might

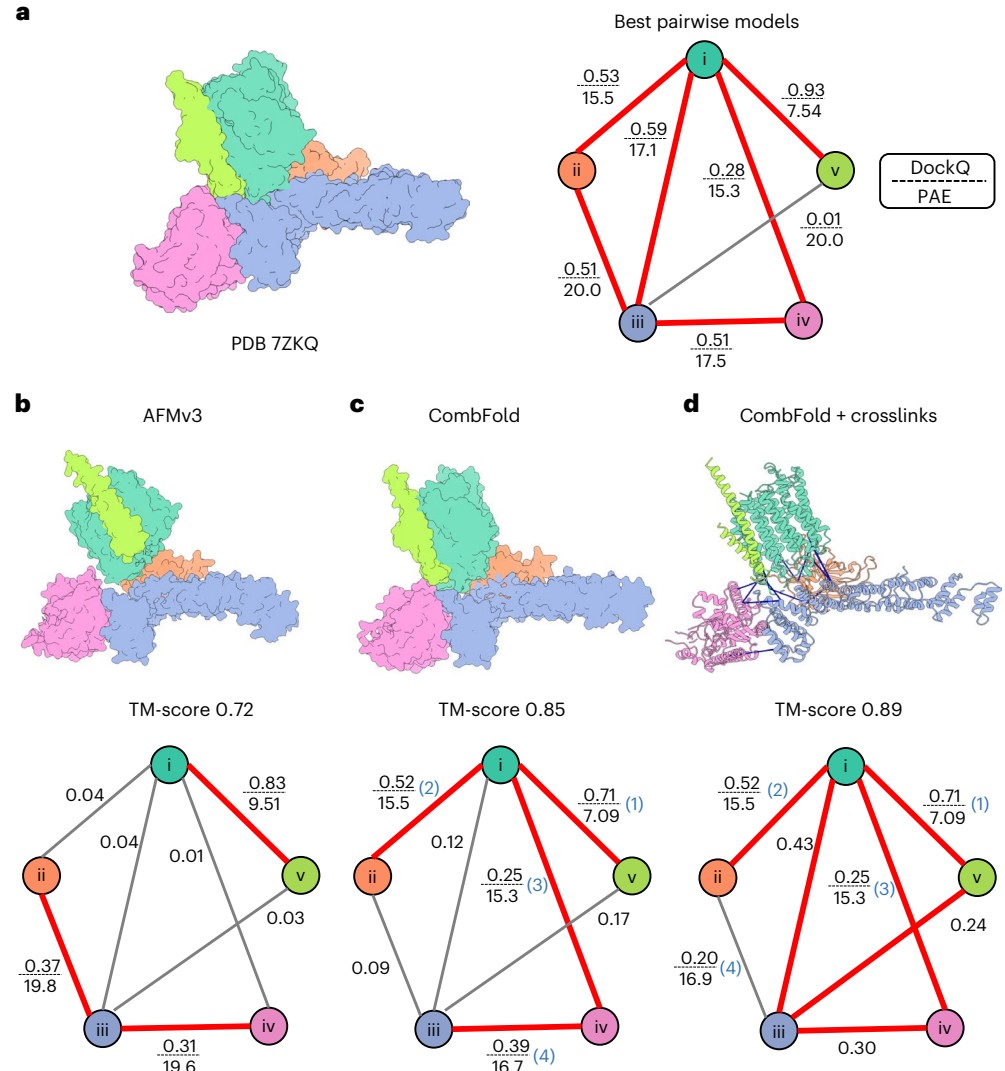

**Fig. 5 | The advantage of hierarchical assembly over global AFM.**
**a**, Experimental structure of the early Pp module assembly intermediate of complex I (left) and the interaction graph (right). The node colors correspond to subunit colors. The edges are shown for all subunit pairs that have close contacting amino acids. Edges are labeled with the highest DockQ generated by AFM in the first stage of CombFold and their average PAE. **b**–**d**, Predicted models (top) with the quality of their pairwise interactions (DockQ, PAE) mapped on the interaction graph (bottom) for AFM (**b**), CombFold (**c**) and CombFold with crosslinks (**d**). Accurate pairwise interactions (DockQ >0.23) are in red. Crosslinks are shown as blue lines. CombFold assembly order is indicated on the graph with numbers in parentheses (blue).

be obtained from pairwise docking methods[28,31,60] as in the original CombDock method[17]. This will enable us to further increase the success rate of our method.

We compare CombFold to other complex structure prediction methods. Docking-based methods such as HADDOCK[55] are unable to predict large complexes[35] (Extended Data Fig. 3c). When compared to the Monte-Carlo Tree Search assembly (MoLPC) that is mainly applicable to homomeric complexes, our combinatorial algorithm doubles the success rate from ~30% to ~60% (Fig. 3a,b). This improvement is particularly significant for heteromeric complexes, where the larger number of subunit combinations leads to an increased number of pairwise interactions. The superior performance of CombFold compared to MoLPC can be attributed to several factors. First, by employing a more exhaustive combinatorial assembly algorithm, and implementing clustering during assembly, we are able to better enumerate the many possible interactions between subunits, resulting in a higher number of accurate assemblies. Second, the enumeration process of CombFold is more strongly based on the confidence score of each transformation, which correlates with accuracy (Extended Data Fig. 7a–e), and

therefore, CombFold is able to select the more confident and accurately predicted interactions (Extended Data Fig. 7f). Third, the usage of a unified representation results in each subunit model being the most confident AFM-generated model of this subunit, which results in an overall more accurate complex structure. Lastly, implementation details such as a more relaxed steric clashes filtering stage, and AFM prediction for groups of more than three subunits efficiently, can be more effective when implementing assembly-based methods.

We also compare CombFold to end-to-end AFM, which is considered state of the art for predicting entire complexes. We find that AFM is still limited compared to assembly methods by the maximal total length of the complex and lack of diversity in the generated structures. Most complexes that are accurately predicted by AFM are also accurately assembled by CombFold based on the pairwise interactions from AFM (Fig. 2d). Two primary reasons account for CombFold's superior performance compared to AFM. First, the stage that generates pairwise subunit interactions enables us to find a higher number of accurately predicted pairs. For example, for the early Pp module assembly intermediate of complex I, we find six pairwise interactions of acceptable quality (DockQ

>0.23, Fig. 5a). As a result in the assembly stage, several assembly pathways are possible because only four pairwise interactions that produce a spanning tree of all subunits are needed to assemble the complex. In contrast, AFM applied on the whole complex correctly predicts only three pairwise interactions (Fig. 5b). Second, even if the pairwise interaction was not predicted correctly by AFM, it can still form during the assembly process (Fig. 5d, subunits iii–v). This also applies to other end-to-end (single step) methods, such as RosettaFold2 and AlphaLink.

While some complexes assemble into stable structures, others are dynamic and exist in multiple states. The heterogeneity can be both compositional with subunits that interact transiently or conformational with flexible proteins or a combination of both[61]. Addressing this heterogeneity is still challenging. For example, compositional heterogeneity can be addressed similarly to stoichiometry by enumerating compositions during assembly. The conformational heterogeneity is currently addressed based on additional structural information, such as cryo-EM[62–64], cryo-electron tomography[65], crosslinking mass spectrometry[66] and single-molecule FRET[67]. The Bayesian approach that can account for most sources of uncertainty in data without overfitting is often used for determining structural ensembles[68]. This approach estimates the probability of a model, given information available about the system, including both prior knowledge and newly acquired experimental data. It was successfully integrated into data-driven MD simulations and adopted for multiple types of data, including cryo-EM density maps[62] and contact or distance information from multiple sources[69]. Our current implementation can integrate distance-based information into the assembly stage and generate multiple models that are consistent with the data. Moreover, models generated by CombFold can be used as starting points for generating dynamic ensembles using data-driven simulation approaches, such as CryoFold[70,71].

Large datasets of experimentally observed protein–protein interactions and assemblies are available from Complex Portal, Corum and STRING[36,72,73]. In addition, crosslinking mass spectrometry is providing large datasets of interactions[74]. These datasets can be used by Comb-Fold, including crosslinks that can be converted into distance restraints and integrated into the assembly stage. While the major bottleneck in applying assembly methods on these datasets is unknown stoichiometry, we demonstrate that our approach can be extended to enumerate stoichiometries (Fig. 4) and we plan to further develop this capability to enable the assembly of complexes without known stoichiometry.

## Online content

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

## Methods

### CombFold method

**Definition of subunits.** A subunit is a sequence that can be either an independent chain of the complex or a part of a chain (for example, a certain domain). Sometimes it is necessary to divide a chain into a number of subunits−either because the chain is too long to be predicted by AFM or because domains are connected by a long linker and are not in spatial proximity. In case a chain is too long for modeling with other chains, or if it is known to contain a long inter-domain linker, it is best to divide it into structural domains based on predicted disordered regions using tools, such as IUPred3 (ref. 75).

In Benchmark 4, each subunit was defined as a single chain according to definitions supplied by CASP. The two targets that are long single chains (T1165 and T1169) were divided into subunits according to IUPred3 (ref. 75). The predicted disordered regions connecting the domains were not included in the prediction. In all other benchmarks a full chain was used as a subunit as defined in the SEQRES segment of the PDB entry for almost all cases. Due to a high number of long chains in Benchmarks 2 and 3, we opted for a simple split procedure without relying on predicted disorder regions. In Benchmark 2, long chains in five complexes (PDBs 8HIL, 8F50, 8ADL, 8A3T and 7OZN) were divided into subunits evenly until every subunit pair could be predicted by AFMv3. In Benchmark 3, long chains in two complexes (PDBs 1I50 and 6KWY) were divided into two subunits, one containing the first 1,000 amino acids and the other with the rest. For Complex Portal predictions, the UniProt sequences were divided similarly to Benchmark 2.

**AlphaFold2 structure prediction.** In the first stage, we run AFM on each possible pairing of the subunits. Proteins, both homomers and heteromers, have the ability to create intertwined structures where the interacting chains exchange small segments or compact protein substructures. These interactions can result in a wide range of quaternary arrangements, including dimers, or higher-order oligomers[76]. To account for this, AFM prediction is applied for larger subsets of three to five subunits as follows. For each subunit, we select the most likely interacting subunits based on the pairwise PAE interaction score and use them to build larger subsets (Methods). Here we limit our calculations to the total length of input sequences of 1,800, which can be run on standard GPUs.

AlphaFold2 runs were performed using ColabFold[77] with default parameters (without templates), producing five structures per run. Subunits were inputted as separate chains. For Benchmarks 1 and 3, we used AFMv2 and AlphaFold-ptm to obtain ten structural models. For comparison to CombFold on Benchmark 1, only end-to-end AFMv2 was used. For Benchmark 2, CASP15 and Complex Portal predictions, we used AFMv3 only, as it was not trained on these targets. CombFold predictions on Benchmark 2 were compared to end-to-end AFMv3.

**Extracting representative subunit structures.** Each subunit structure from AFM predictions is ranked on the basis of the mean plDDT score using all predicted structures from AFM runs for pairs and larger subsets. The structure with the maximal score is selected as the 'representative subunit structure' for the assembly stage. Additional criteria were examined as possible ranking scores including the average PAE score for the structure, the maximal plDDT or the interaction score with other subunits in AFM prediction. There were no significant differences between the described possibilities; the mean plDDT, which is easy to calculate and more widely used, was chosen.

**Computing pairwise transformations.** The method computes for each pair of subunits a list of possible transformations between them based on their interaction models from AlphaFold2. All pairs of subunits are extracted from multisubunit predictions. For each pair, if it is interacting

(Cα−Cα distance <8 Å), the transformation between the subunits is calculated. We can mark the predicted interacting structure for two subunits $A$ and $B$, and two representative structures for those subunits $A'$ and $B'$. Notice that even though $A$ and $A'$ are the same molecules, the different interactions in each AFM model will result in different structures and different reference frames for $A$ and $A'$. We would like to calculate a transformation between the representatives $B'$ to $A'$ that will result in the interaction interface as close as possible to that of the examined model pair $A$ and $B$. To achieve this, the transformation $T_1$ that aligns $A'$ on $A$ is calculated by computing the transformation that minimizes root mean square deviation (RMSD)[78,79]. Similarly, the transformation $T_2$ that aligns $B'$ on $B$ is calculated. Finally, the desired transformation is composed as $T_2 \circ T_1^{-1}$. A problem arises when a subunit has a disordered region−this region will be folded differently in each predicted model, which can substantially affect the alignment and the resulting transformation. Therefore, during the alignment, we consider only amino acids that have a high plDDT score (>80) or at least half of the amino acids with the highest plDDT.

Each transformation is scored using the PAE score of the two subunits. PAE score is computed by AFM for any two amino acids in the structure, predicting their alignment error relating to each other. The PAE score values are between 0 and 30, with lower values corresponding to a lower predicted error. The transformation score is calculated and normalized to be between 1 and 100 by the equation $\max\{1, 100 - P^2/4\}$ where $P$ is the average value of PAE of the two interacting subunits. This expression gives the score quadratic properties so that small differences in low $P$ scores (which are usually at least 1) will be meaningful, while for high $P$ scores, there is not much difference between the score of transformations as it is predicted to be inaccurate.

Multiple possibilities for scoring were considered, including PAE, the minimal PAE, the interface PAE of the interacting amino acids only, interface predicted TM-score (ipTM) and interface plDDT (ipLDDT), which is widely used[12,35,80]. All scores had a comparable correlation with Cα RMSD (Pearson $r$ of ~0.5−0.6, Extended Data Fig. 7a−e). The advantage of our PAE-based score is that incorrect interfaces consistently have low scores (Extended Data Fig. 7e). Our analysis of average PAE distributions of all AFM pairwise interaction modes versus the ones that were selected for top-1 assembly models revealed that CombFold indeed selects the interactions with lower PAE scores (Extended Data Fig. 7f).

### Combinatorial assembly of subunits

The input to the assembly stage is a list of representative structures of subunits and a list of pairwise transformations between subunits. The output is a list of assembled complexes containing all the subunits. If all the subunits can not be assembled, the algorithm outputs partial complexes containing the largest number of input subunits. The assembly algorithm proceeds with $N$ iterations, where $N$ is the number of input subunits. In each iteration, the size of the subcomplexes created is increased, until the $N$th iteration, where the subcomplexes computed contain all input subunits.

Each iteration contains three stages: subcomplexes expansion, filtering and clustering. The first stage creates new subcomplexes based on smaller subcomplexes from previous iterations and pairwise transformations that were provided to the algorithm. Each new subcomplex is scored on the basis of the scores of the pairwise transformations that were used to generate it. The second stage filters assembled subcomplexes with steric clashes between subunits. The third stage clusters subcomplexes with the same subunit composition and saves $K$ best-scoring subcomplexes. Optionally, the final structures can be relaxed to resolve steric clashes.

**Expansion stage.** In this stage, we attempt to connect pairs of subcomplexes that have no overlapping subunits and with the total number of $i$ subunits, where $i$ is the iteration number. For each pair of subunits in

the two subcomplexes (of sizes $k$ and $i − k$), a new larger subcomplex is computed for each input pairwise transformation between those subunits. The transformation is applied to all the subunits of the second subcomplex, thus bringing it to the first subcomplex.

There is a special reward for scoring symmetrical subcomplexes with over five identical subunits transformed with the same pairwise subunit transformation. This reward compensates for the assembly being based on pairwise subunit interactions, compared to the full assembly by AFM, which is likely to result in lower PAE scores if a symmetrical structure was formed. Therefore, if a symmetric structure was generated on the basis of pairwise subunit transformations, the new score is calculated as $(S + S \times (100 − S)/100)$, where $S$ is the original score of the transformation.

**Filtering stage.** As the pairwise transformations can be at least partially inaccurate, applying some of them can result in subcomplexes with steric clashes or violated distance constraints and restraints. Steric clashes are checked for all backbone atoms with plDDT higher than 80 because the representative structures can contain disordered regions, which are likely to clash with other subunits as they are left static during the assembly (Extended Data Fig. 1). A backbone atom of one subunit is considered as clashing if its center penetrates by more than 1 Å into the surface of another subunit. The steric clash test is performed for all pairs of subunits, one from each subcomplex. A subcomplex is filtered if there are over 5% of a subunit's backbone atoms clashing with another subunit.

Distance constraints are imposed on different subunits from the same chain to enforce sequence connectivity. A subcomplex is discarded if the distance between consecutive amino acids from two subunits is greater than the number of linker amino acids multiplied by 3 Å.

**Clustering stage.** RMSD clustering is performed to cluster subcomplexes containing the same subunits. We have used iterative clustering, starting from the best-scoring subcomplex with the RMSD threshold of 1 Å. However, a default RMSD calculation does not account for multiple copies of the same subunit. This means that for a subcomplex with $p$ copies of identical subunits, there will be $p!$ equivalent subcomplexes. In this case, to compare the two subcomplexes we need to find the correspondence between copies of subunits from different subcomplexes that minimizes the RMSD. Incorrect correspondence will lead to high RMSD for similar subcomplexes. To avoid the enumeration of $p!$ configurations, we implemented a heuristic that superimposes only the centroids of the subunits using starting order subunit correspondence. After the initial superimposition, the correspondence for each pair of identical subunits is swapped and the RMSD is recalculated using centroids. If the RMSD has decreased, we proceed with the new correspondence. The swap process is repeated until there is no further RMSD decrease. The final correspondence between subunits is used to calculate the Cα RMSD between the two subcomplexes.

After clustering, only the $K$ best-scored subcomplexes of size $i$ will be saved for the next iteration (on the presented benchmarks $K = 100$). Clustering aids in diversifying the stored subcomplexes and avoiding the dominance of suboptimal ones in the set of subcomplexes for the next iteration.

**Relaxation.** As a result of using representative subunit structures, CombFold may produce structures with steric clashes in interfaces, mainly in side-chains. Therefore, it is recommended to perform an extra step of relaxation of the structure by gradient descent using the Amber[81] force field similar to AlphaFold. This step substantially reduces the clashscore calculated by Molprobity[82] (Extended Data Fig. 3g) while not affecting the structure considerably (change in Cα RMSD <1 Å in all targets of Benchmark 2).

**Data integration.** To consider known interactions between subunits, we group the input subunits into subcomplexes based on the data. Each such group will be assembled separately, followed by the assembly of the groups and remaining subunits into a larger complex. Therefore, the information is used to enforce a specific assembly order that is consistent with the known interactions.

The crosslinking mass spectrometry information is converted into distance restraints. A restraint is considered satisfied if the Cα–Cα distance is below a distance threshold. The threshold is defined by the user on the basis of the length of the crosslinker. In the case of ambiguity of crosslinked residues due to multiple copies of the same subunit, we require that one of the possible distances restrained by the crosslink is below the distance threshold. CombFold accounts for the uncertainty in the crosslinking data and in the subunit structures as follows. The uncertainty in the data is accounted for by weighting each crosslink according to its confidence based on the experimental evidence ($w_1$), such as the false discovery rate[83]. To account for uncertainty in the subunit structures, each crosslink is weighted by the average AFM plDDT score of the two crosslinked amino acids ($w_2$). The satisfaction ratio of a subcomplex is calculated as the sum of weights of satisfied distance restraints divided by the sum of weights of all restraints within the given subcomplex (equation (1)). The score of each subcomplex is multiplied by the satisfaction ratio. Consequently, as more restraints are fulfilled, the score increases, making it more probable for the subcomplex to avoid being filtered. A subcomplex is also filtered in the filtering stage if it violates some minimal percentage of its restraints (default 10%).

$$\text{satisfaction ratio} = \frac{\sum_{\text{satisfied}} w_1 \times w_2}{\sum_{\text{all}} w_1 \times w_2} \tag{1}$$

**Predicted confidence.** CombFold predicts the confidence of the assembled structure as a weighted score of the pairwise transformation scores ($S_T$) used in the assembly stage. To calculate the weight of a given transformation ($W_T$), we split the complex into two subcomplexes using the transformation and the complex assembly tree. The weight of the transformation is the number of amino acids in the smaller subcomplex. The idea is that some transformations have a larger effect on the final global structure of the complex, as they affect a larger number of amino acids. The final score is normalized by the total weight of all the transformations used in the assembly stage (equation (2)).

$$\text{predicted confidence} = \frac{\sum_T W_T \times S_T}{\sum_T W_T} \tag{2}$$

## Performance analysis

**Runtimes.** CombFold runtime is dominated by the AFM prediction runs for subunit pairs and larger subsets. On Benchmark 1, the average GPU time for AFM predictions was 709 and 1,429 s for subunit pairs and larger subsets, respectively, running on NVIDIA A30 with 24 GB of memory. However, since our method requires $O(N^2)$ AFM predictions for pairs and $O(N)$ AFM predictions for larger subsets the average total GPU time per complex was 7,093 and 15,404 s for subunit pairs and larger subsets, respectively. It is also important to note that the first stage of CombFold that performs AFM calculations can be trivially distributed into the shorter AFM jobs that can run in parallel. In comparison, the average GPU runtime required for AFM for end-to-end modeling of an entire complex was 5,154 s running on the NVIDIA RTX A6000 with 48 GB of memory ($n = 17$, only cases where AFM was able to produce models were considered, Extended Data Fig. 8). It is important to note that the CombFold runtime is higher for heteromeric complexes containing more unique chains compared to homomeric complexes of similar size, as multiple identical copies of a subunit will use the same AFM interaction models. Benchmark 1 is designed to contain heteromeric complexes with many unique chains; homomeric complexes, such as in Benchmark 3, have lower runtimes.

For example, a symmetrical structure with ten identical chains requires much less GPU time in CombFold compared to naive end-to-end AFM (as we only need to run a job for two copies of the chains which is much faster compared to ten copies). The runtime of the unified representation and combinatorial assembly stages is negligible compared to the AFM and is on average 80–600 s on the different benchmarks on a single central processing unit. In contrast to the generation of pairwise subunit interactions stage, the assembly stage is faster for heteromeric complexes with a higher number of unique chains. The assembly time is much faster compared to MoLPC, where the reported average assembly stage takes 13,000 s.

**Pairwise connectivity.** Given a set of pairwise transformations and a target complex structure, this metric measures how many of the pairwise transformations between subunits from the target complex are present in the set. A graph is built, where each node is a subunit in the target complex and an edge is present if there exists a transformation in the set between those subunits for which the DockQ (ref. 84) score relative to the transformation in the target complex is at an acceptable level (DockQ >0.23). We calculate the connected components of this graph. The pairwise connectivity ratio is defined as the ratio between the number of amino acids in the largest connected component and the total number of amino acids in the complex. A single connected component in the graph (pairwise connectivity 1.0) indicates that there are pairwise transformations that can lead to the assembly of the complex. In contrast, multiple connected components indicate that accurate assembly is not possible with available transformations.

**Comparison to HADDOCK, AlphaLink and RosettaFold2.** HADDOCK and AlphaLink were tested using the simulated crosslinks for Benchmark 2. For HADDOCK (v2.4 with CNSv1.3) the input subunits were the same representative subunits that were used for CombFold assembly. For AlphaLink (v2.2), a model that was trained on restraints with an upper bound of 25 Å on the Cα-Cα distances was used. RosettaFold2 was tested using RF_apr23 model weights on Benchmark 2 without crosslinks.

**Comparison to MoLPC.** MoLPC evaluation used a TM-score above 0.8 to define a high-quality prediction. Here we use the same definition of high-quality prediction. We find that a prediction with a TM-score of 0.7 can have a correct global shape (Figs. 3h and 5b). Therefore, we define an additional acceptable-quality category for predictions with a TM-score above 0.7. In the original MoLPC publication, the success rate was calculated as a fraction of benchmark cases with a high-quality prediction out of cases where at least one assembly was obtained. Note that MoLPC was able to obtain some predictions for 91 out of 175 Benchmark 3 cases. Here we define a success rate as a fraction of benchmark cases with an acceptable-quality prediction out of all benchmark cases. In addition, while MoLPC has presented separate success rates for AFM-based or FoldDock-based pipelines, we have considered results from both pipelines in our calculated success rate. We recalculated the success rate of MoLPC according to our definitions, resulting in slightly different values.

**Visualizations.** Protein complexes were visualized using ChimeraX (ref. 85). Graphs were created using Matplotlib[86].

### Reporting summary

Further information on research design is available in the Nature Portfolio Reporting Summary linked to this article.

## Data availability

The PDB codes for Benchmarks 1–3, scripts and data for manuscript figures are part of the repository https://github.com/dina-lab3D/CombFold.

## Code availability

CombFold assembly is implemented using C++. The code, Colab notebook, and tutorial for CombFold are available at https://github.com/dina-lab3D/CombFold. There is also a Code Ocean capsule available for running the assembly algorithm at https://codeocean.com/capsule/8791899.

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

## Acknowledgements

D.S.-D. and B.S. are supported by the Israeli Science Foundation (ISF 1466/18), NIH NIAID (R01AI163011-01A1) and Minerva Stiftung. The funders had no role in study design, data collection and analysis, decision to publish, or preparation of the manuscript. Molecular graphics and analyses were performed with UCSF ChimeraX, developed by the Resource for Biocomputing, Visualization, and Informatics at the University of California, San Francisco, with support from National Institutes of Health R01-GM129325 and the Office of Cyber Infrastructure and Computational Biology, National Institute of Allergy and Infectious Diseases.

## Author contributions

Conceptualization was carried out by B.S. and D.S.-D. The methodology, software development, investigation, data curation, benchmarking, validation and visualization were developed by B.S. Supervision and project administration were carried out by D.S.-D. Writing of the paper was done by B.S. and D.S.-D.

## Competing interests

The authors declare no competing interests.

## Additional information

**Extended data** is available for this paper at https://doi.org/10.1038/s41592-024-02174-0.

**Correspondence and requests for materials** should be addressed to Dina Schneidman-Duhovny.

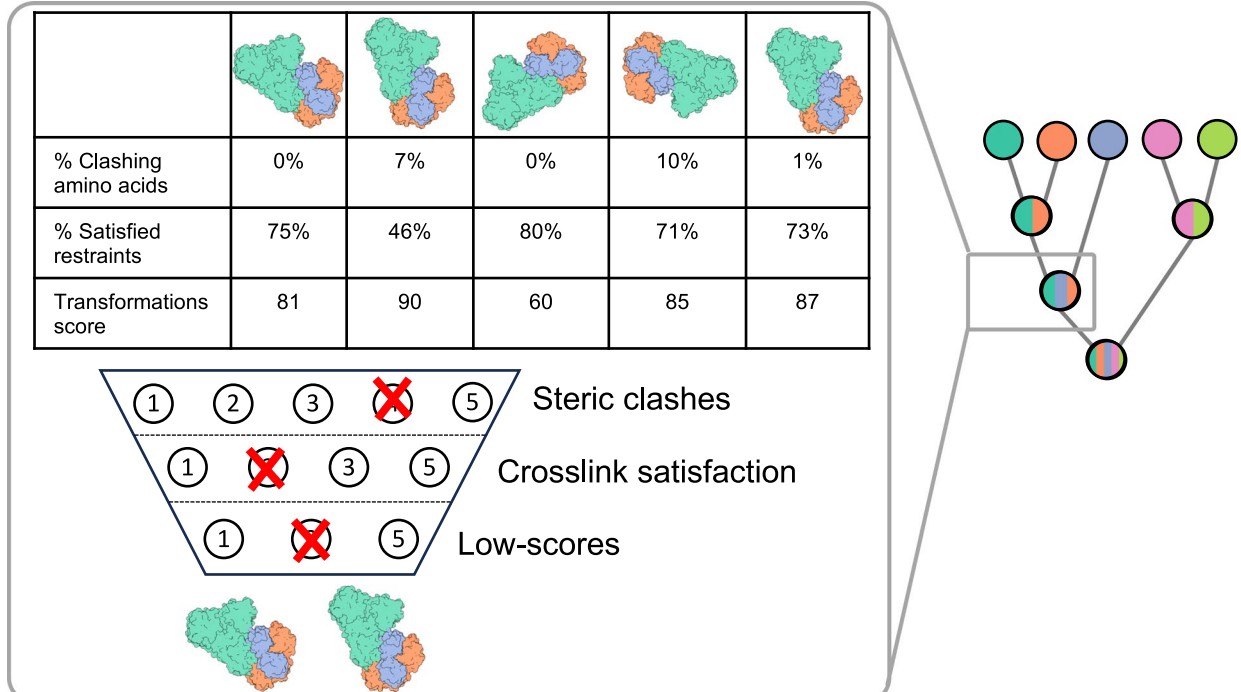

| | | | | | |
|---|---|---|---|---|---|
| % Clashing amino acids | 0% | 7% | 0% | 10% | 1% |
| % Satisfied restraints | 75% | 46% | 80% | 71% | 73% |
| Transformations score | 81 | 90 | 60 | 85 | 87 |

**Extended Data Fig. 1 | CombFold filtering visualization.** For each assembly tree, in each step, CombFold joins two previously assembled subcomplexes, into many new subcomplexes by applying input transformations between pairs of subunits. These new subcomplexes are filtered to discard suboptimal subcomplexes. The first filter is by crossing a threshold of allowed steric clashes between amino acids of different subunits, in this example, the threshold is 5%. The second filter is by not satisfying enough of the distance restraints present in the subcomplex, here the threshold is 70%. The last filter scores each subcomplex based on the used transformation scores and the distance restraints satisfaction rate.

a

b

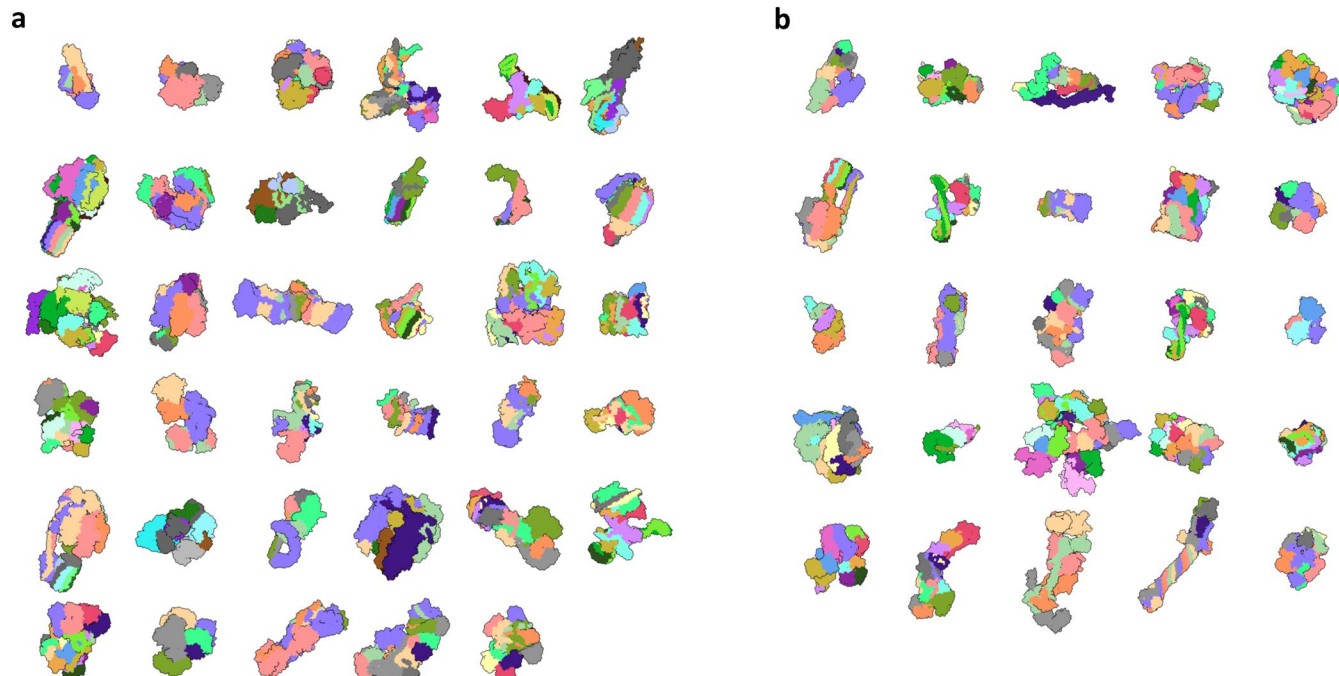

**Extended Data Fig. 2 | Heteromeric benchmark datasets.** Heteromeric complexes (colored by chain) from (**a**) Benchmark 1 and (**b**) Benchmark 2.

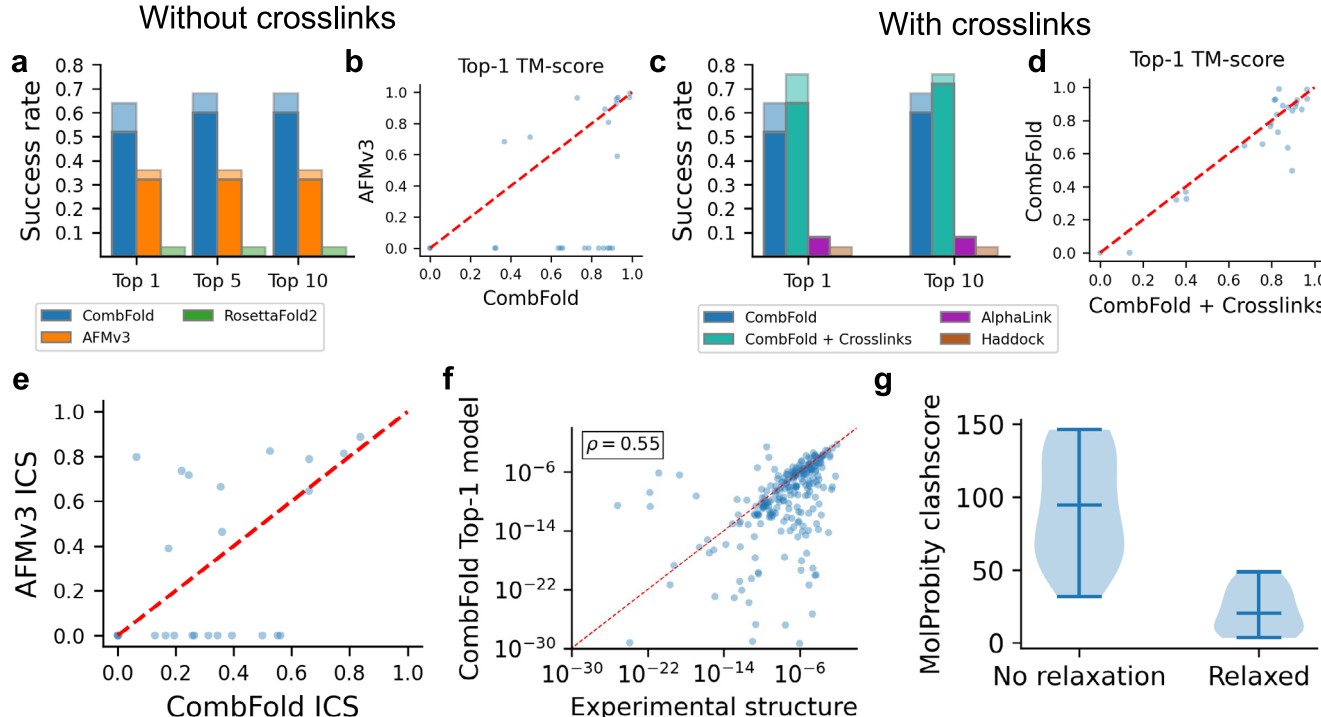

**Extended Data Fig. 3 | Accuracy of CombFold on Benchmark 2. (a)** The Top-N (N = 1, 5, 10) success rate of CombFold (blue), AFMv3 (orange), and RosettaFold2 (green). **(b)** TM-score of AFMv3 models vs. CombFold models for Top-5 results **(c)** The Top-N (N = 1, 5, 10) success rate of CombFold (blue),CombFold with crosslinks (turquoise), AlphaLink (purple) and HADDOCK(brown). **(d)** TM-score of CombFold models with crosslinks vs. without crosslinks for Top-1 results. **(e)** Interface contact similarity (ICS) of CombFold vs. AFMv3 for Top-1 model.

**(f)** Comparison of PRODIGY predicted dissociation constants for interfaces of experimental structures vs. interfaces of structure models generated by CombFold. Spearman correlation of 0.55. **(g)** Distributions of clashscores are calculated using MolProbity for interfaces in the models of CombFold output models (left, N = 17) and the same models after relaxation (right, N = 17). Error bars indicate maxima, mean, and minima from top to bottom respectively.

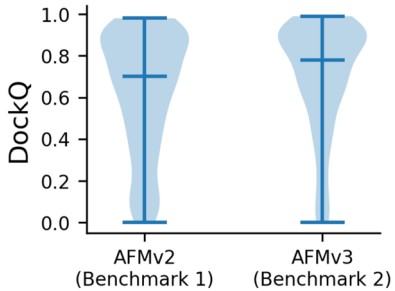

**Extended Data Fig. 4 | Accuracy of pairwise predictions for AFMv2 and AFMv3.** DockQ scores of pairwise interactions predicted by AFM on Benchmark 1 (AFMv2, N = 469) and Benchmark 2 (AFMv3, N = 445), for which the PAE-based score is over 50. The median score is 0.70 and 0.78 for AFMv2 and AFMv3, respectively. Error bars indicate maxima, mean, and minima from top to bottom respectively.

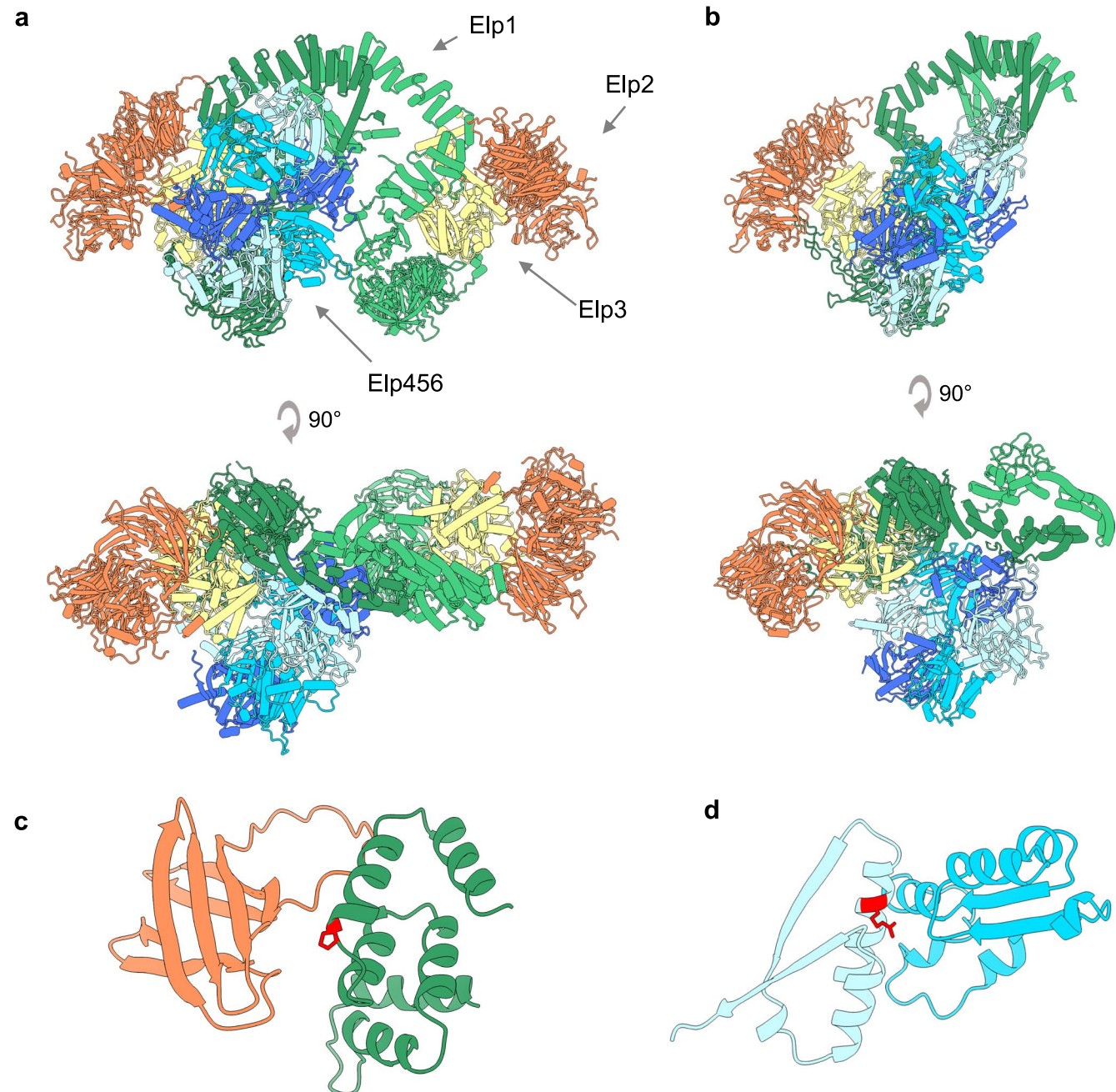

**Extended Data Fig. 5 | Modeling the human Elongator holoenzyme complex.**
(**a**) CombFold prediction for the human Elongator holoenzyme complex.
(**b**) Part of the complex structure in yeast, as determined by Cryo-EM (PDB 8ASV).
(**c**) The interface between Elp1 (green) and Elp2 (orange) with a likely pathogenic mutation P914L in Elp1 is depicted as sticks (red). (**d**) The interface between Elp4 (light blue) and Elp6 (sky blue) with a pathogenic mutation R289W in Elp4 is depicted as sticks (red).

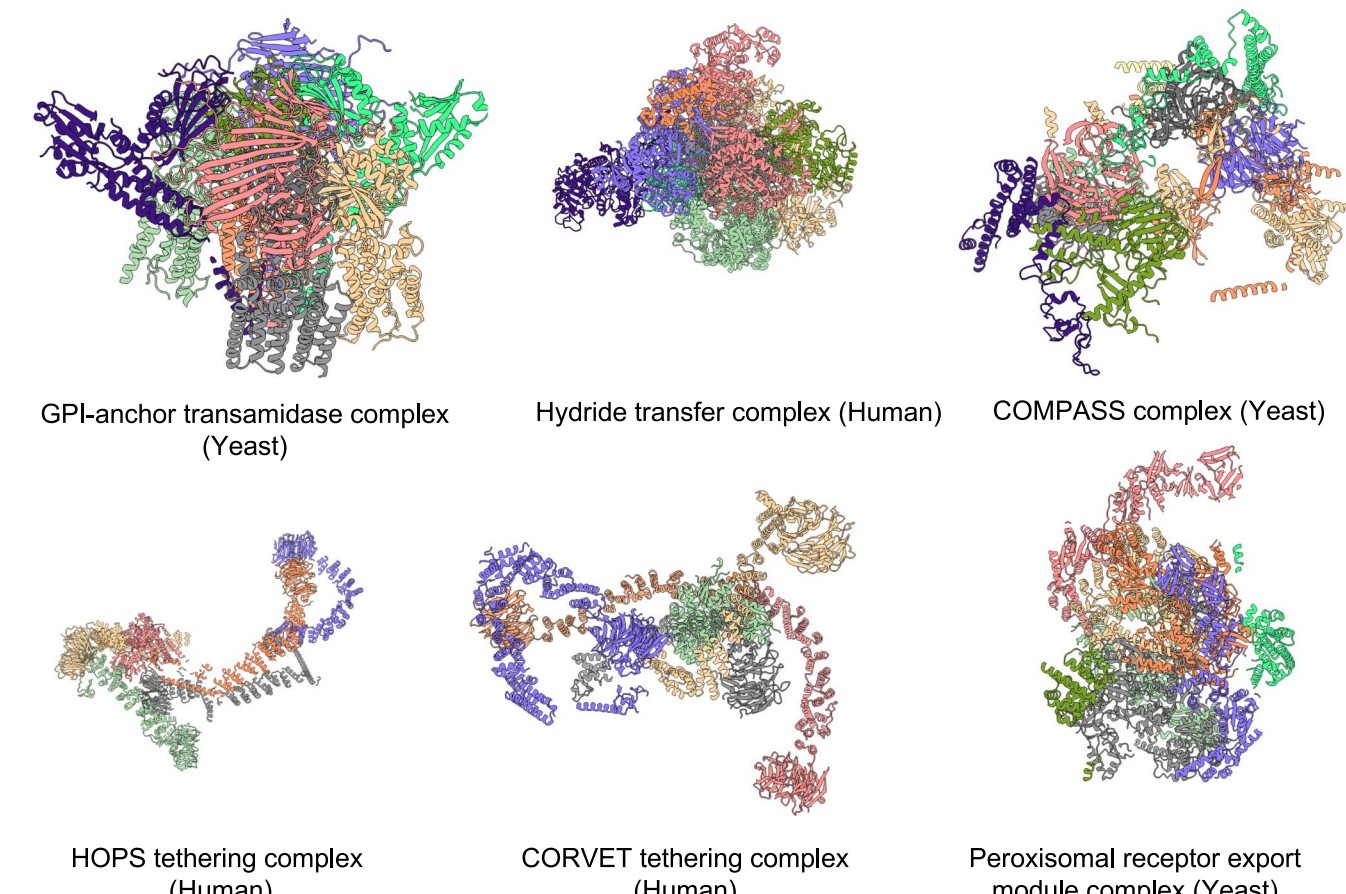

GPI-anchor transamidase complex
(Yeast)

Hydride transfer complex (Human)

COMPASS complex (Yeast)

HOPS tethering complex
(Human)

CORVET tethering complex
(Human)

Peroxisomal receptor export
module complex (Yeast)

**Extended Data Fig. 6 | Complex Portal Predicted Complexes.** Predicted complexes from Complex Portal with High or Medium confidence.

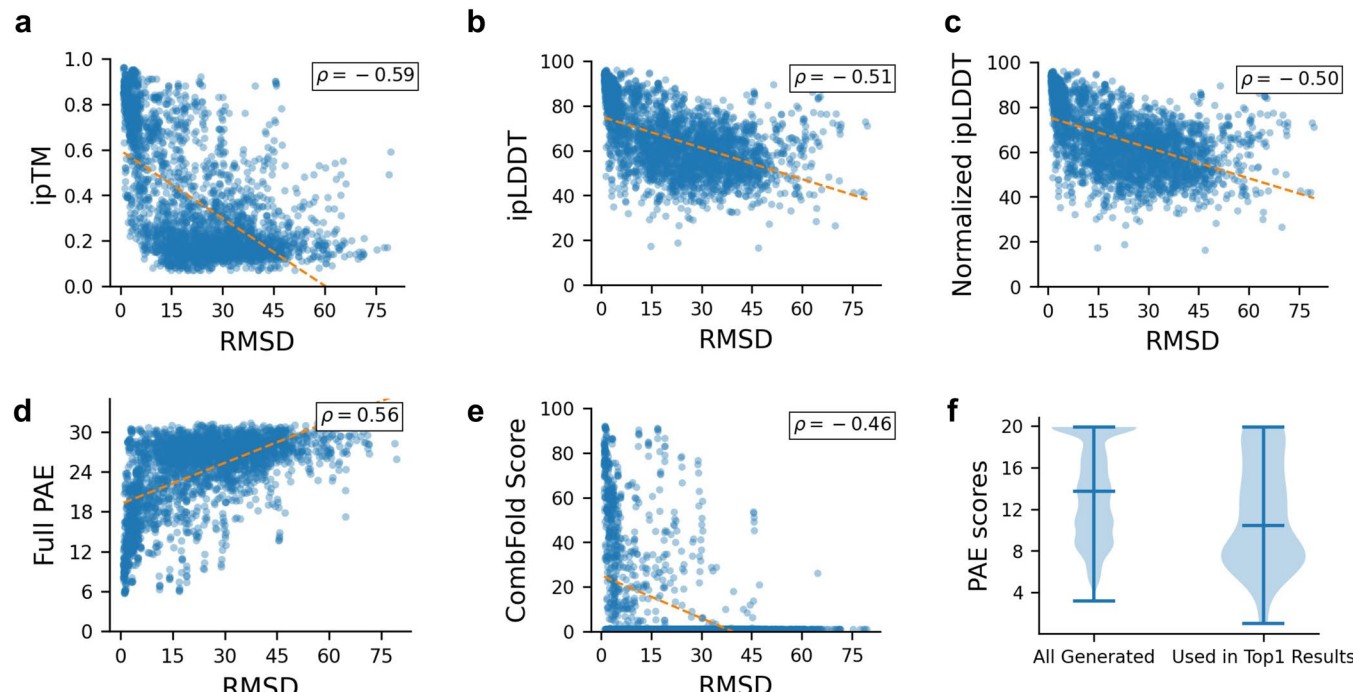

**Extended Data Fig. 7 | Analysis of pairwise scoring functions.** Each graph (**a**)-(**e**) presents a scoring function vs. pairwise RMSD, calculated on Benchmark 2. (**a**) ipTM. (**b**) iplDDT - for each interface the average plDDT of its amino acids is calculated and those averages are averaged. (**c**) iplDDT, where the plDDT of each interface is weighted by its size (**d**) the average PAE scores of all amino acids in the pair (**e**) CombFold score, based on PAE as described in Methods. (**f**) The distribution of average PAE scores for all generated pairwise interactions (left, N = 34,365) vs. the distribution of PAE scores in Top-1 models (right, N = 310) created by CombFold. Error bars indicate maxima, mean, and minima from top to bottom respectively.

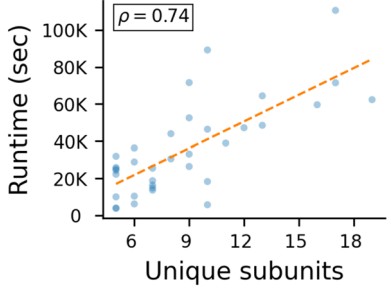

**Extended Data Fig. 8 | Runtime Analysis.** CombFold runtime vs. the number of unique subunits. Calculated on all cases in Benchmark 1. Pearson correlation of 0.74.

# Reporting Summary

## Statistics

For all statistical analyses, confirm that the following items are present in the figure legend, table legend, main text, or Methods section.

| n/a | Confirmed | |
|---|---|---|
| ☐ | ☒ | The exact sample size (*n*) for each experimental group/condition, given as a discrete number and unit of measurement |
| ☒ | ☐ | A statement on whether measurements were taken from distinct samples or whether the same sample was measured repeatedly |
| ☒ | ☐ | The statistical test(s) used AND whether they are one- or two-sided<br>*Only common tests should be described solely by name; describe more complex techniques in the Methods section.* |
| ☒ | ☐ | A description of all covariates tested |
| ☒ | ☐ | A description of any assumptions or corrections, such as tests of normality and adjustment for multiple comparisons |
| ☐ | ☒ | A full description of the statistical parameters including central tendency (e.g. means) or other basic estimates (e.g. regression coefficient) AND variation (e.g. standard deviation) or associated estimates of uncertainty (e.g. confidence intervals) |
| ☒ | ☐ | For null hypothesis testing, the test statistic (e.g. *F*, *t*, *r*) with confidence intervals, effect sizes, degrees of freedom and *P* value noted<br>*Give P values as exact values whenever suitable.* |
| ☒ | ☐ | For Bayesian analysis, information on the choice of priors and Markov chain Monte Carlo settings |
| ☒ | ☐ | For hierarchical and complex designs, identification of the appropriate level for tests and full reporting of outcomes |
| ☐ | ☒ | Estimates of effect sizes (e.g. Cohen's *d*, Pearson's *r*), indicating how they were calculated |

*Our web collection on statistics for biologists contains articles on many of the points above.*

## Software and code

Policy information about availability of computer code

| Data collection | No Software used for data collection. |
|---|---|
| Data analysis | We have used the following software for data analysis:<br>1. TM-scores were calculated using MMalign (2021-08-16)<br>2. Biopython 1.79 was used for RMSD calculations<br>3. The plots and correlations were calculated using python3 matplotlib 3.53 versions<br>4. The structures were analyzed and visualized using ChimeraX 1.5<br>5. Amber calculations completed using implementation alphafold python package<br><br>CombFold assembly is implemented using C++.  The code, Colab notebook, and tutorial for CombFold are available In GitHub: https://github.com/dina-lab3D/CombFold<br><br>Project Code Ocean Capsule:<br>https://codeocean.com/capsule/8791899 |

For manuscripts utilizing custom algorithms or software that are central to the research but not yet described in published literature, software must be made available to editors and reviewers. We strongly encourage code deposition in a community repository (e.g. GitHub). See the Nature Portfolio guidelines for submitting code & software for further information.

## Data

Policy information about availability of data

All manuscripts must include a data availability statement. This statement should provide the following information, where applicable:

- Accession codes, unique identifiers, or web links for publicly available datasets
- A description of any restrictions on data availability
- For clinical datasets or third party data, please ensure that the statement adheres to our policy

In our study we rely on the PDB structures. The PDB codes are available in the github repository.
Scripts and data for manuscript figures are part of the repository. The PDB codes for Benchmarks 1-3 are also part of the repository:
https://github.com/dina-lab3D/CombFold/tree/master/paper_resources/datasets

## Human research participants

Policy information about studies involving human research participants and Sex and Gender in Research.

| | |
|---|---|
| Reporting on sex and gender | N/A |
| Population characteristics | N/A |
| Recruitment | N/A |
| Ethics oversight | N/A |

Note that full information on the approval of the study protocol must also be provided in the manuscript.

# Field-specific reporting

Please select the one below that is the best fit for your research. If you are not sure, read the appropriate sections before making your selection.

☒ Life sciences　　　☐ Behavioural & social sciences　　　☐ Ecological, evolutionary & environmental sciences

For a reference copy of the document with all sections, see nature.com/documents/nr-reporting-summary-flat.pdf

# Life sciences study design

All studies must disclose on these points even when the disclosure is negative.

| | |
|---|---|
| Sample size | We have included in our Benchmarks all the available PDB structures that passed our criteria as detailed in the Data exclusions. |
| Data exclusions | We have excluded from our Benchmarks all the structures that AlphaFold2 or AlphaFold-multimer was trained on based on the release date and sequence identity cut-offs as described in Methods. This exclusion is required as those structures are expected to be easier to solve and therefore will not be an accurate way to measure our method. |
| Replication | N/A as there are no experiments in the study and all computations are deterministic and reproducible. |
| Randomization | N/A as there are no experiments in the study and all computations are deterministic and reproducible. |
| Blinding | N/A as there are no experiments in the study and all computations are deterministic and reproducible. |

# Reporting for specific materials, systems and methods

We require information from authors about some types of materials, experimental systems and methods used in many studies. Here, indicate whether each material, system or method listed is relevant to your study. If you are not sure if a list item applies to your research, read the appropriate section before selecting a response.

## Materials & experimental systems

| n/a | Involved in the study |
|-----|----------------------|
| ☒ | ☐ Antibodies |
| ☒ | ☐ Eukaryotic cell lines |
| ☒ | ☐ Palaeontology and archaeology |
| ☒ | ☐ Animals and other organisms |
| ☒ | ☐ Clinical data |
| ☒ | ☐ Dual use research of concern |

## Methods

| n/a | Involved in the study |
|-----|----------------------|
| ☒ | ☐ ChIP-seq |
| ☒ | ☐ Flow cytometry |
| ☒ | ☐ MRI-based neuroimaging |

