## [Peer Review File · Nature Methods]

Peer Review Information

Manuscript Title: Predicting structures of large protein assemblies using combinatorial assembly algorithm and AlphaFold2

Corresponding author name(s): Dina Schneidman

Editorial Notes: None

Reviewer Comments & Decisions:

Decision Letter, initial version:

Dear Dina,

Your Article, "Predicting structures of large protein assemblies using combinatorial assembly algorithm and AlphaFold2", has now been seen by 3 reviewers. As you will see from their comments below, although the reviewers find your work of considerable potential interest, they have raised a number of concerns. We are interested in the possibility of publishing your paper in Nature Methods, but would like to consider your response to these concerns before we reach a final decision on publication.

We therefore invite you to revise your manuscript to address these concerns. In addition to all the technical and data presentation concerns that the reviewers have, we think it will be important to demonstrate that the method offers a clear advance over other similar methods.

* include a point-by-point response to the reviewers and to any editorial suggestions

* please underline/highlight any additions to the text or areas with other significant changes to facilitate review of the revised manuscript

* address the points listed described below to conform to our open science requirements

* ensure it complies with our general format requirements as set out in our guide to authors at www.nature.com/naturemethods

* resubmit all the necessary files electronically by using the link below to access your home page

[Redacted] This URL links to your confidential home page and associated information about manuscripts you may have submitted, or that you are reviewing for us. If you wish to forward this email to co-authors, please delete the link to your homepage.

We hope to receive your revised paper within 10 weeks. If you cannot send it within this time, please let us know. In this event, we will still be happy to reconsider your paper at a later date so long as nothing similar has been accepted for publication at Nature Methods or published elsewhere.

OPEN SCIENCE REQUIREMENTS

REPORTING SUMMARY AND EDITORIAL POLICY CHECKLISTS

Please note that these forms are dynamic 'smart pdfs' and must therefore be downloaded and completed in Adobe Reader. We will then flatten them for ease of use by the reviewers. If you would

like to reference the guidance text as you complete the template, please access these flattened versions at <http://www.nature.com/authors/policies/availability.html>.

DATA AVAILABILITY

We strongly encourage you to deposit all new data associated with the paper in a persistent repository where they can be freely and enduringly accessed. We recommend submitting the data to discipline-specific and community-recognized repositories; a list of repositories is provided here:

<http://www.nature.com/sdata/policies/repositories>

All novel DNA and RNA sequencing data, protein sequences, genetic polymorphisms, linked genotype and phenotype data, gene expression data, macromolecular structures, and proteomics data must be deposited in a publicly accessible database, and accession codes and associated hyperlinks must be provided in the “Data Availability” section.

Please include a “Data availability” subsection in the Online Methods. This section should inform readers about the availability of the data used to support the conclusions of your study, including accession codes to public repositories, references to source data that may be published alongside the paper, unique identifiers such as URLs to data repository entries, or data set DOIs, and any other statement about data availability. At a minimum, you should include the following statement: “The data that support the findings of this study are available from the corresponding author upon request”, describing which data is available upon request and mentioning any restrictions on availability. If DOIs are provided, please include these in the Reference list (authors, title, publisher (repository name), identifier, year). For more guidance on how to write this section please see:

<http://www.nature.com/authors/policies/data/data-availability-statements-data-citations.pdf>

CODE AVAILABILITY

Please include a “Code Availability” subsection in the Online Methods which details how your custom code is made available. Only in rare cases (where code is not central to the main conclusions of the paper) is the statement “available upon request” allowed (and reasons should be specified).

MATERIALS AVAILABILITY

SUPPLEMENTARY PROTOCOL

To help facilitate reproducibility and uptake of your method, we ask you to prepare a step-by-step Supplementary Protocol for the method described in this paper. We [encourage authors to share their step-by-step experimental protocols](https://www.nature.com/nature-research/editorial-policies/reporting-standards#protocols) on a protocol sharing platform of their choice and report the protocol DOI in the reference list. Nature Portfolio's Protocol Exchange is a free-to-use and open resource for protocols; protocols deposited in Protocol Exchange are citable and can be linked from the published article. More details can found at www.nature.com/protocolexchange/about.

ORCID

Sincerely,
Arunima

Arunima Singh, Ph.D.
Senior Editor
Nature Methods

Reviewers' Comments:

Reviewer #1:

Remarks to the Author:

The article reports an AlphaFold-guided hierarchical docking procedure to construct models of protein assemblies, which also benefits from using experimental constraints. Beyond assembly construction, the tool is also used for stoichiometry prediction. The demonstration example on google collab is working. So my questions are primarily focused on the details of the tool.

1. Why is the tool better than AFM - what is making up for the failure of AFM and how ? Take one example and show it through all the steps to make this point.

2. For comparisons purposes what are the other strategies that can be utilized ? Beyond AFM, as an obvious choice comparison with this Rosetta tool (<https://pubmed.ncbi.nlm.nih.gov/33863889/>) should be made. The differences will have to be significant prove that the current work is not incremental.
3. How's is uncertainty in the experimental data accounted for during the integration ? Simple harmonic potentials as constraints will have a number of overfitting issues if the data is not gaussian distributed (which is often the case) - how is this issue addressed ?
4. Biophysical characterization (if available) of the assemblies with unknown structures will be even better. Mutational or biochemical validations are good nonetheless, but qualitative. Since no energy-based metric is optimized for model construction, these can potentially be used for validation. In the case experimental kD values are available for some complexes, the predicted models can be fed into the PRODIGY server and the statistic of the kD values from these CombFold models can be compared with those from the experiments.
5. Molprobilty scores telling about the quality of the produced models particularly at the interfaces will be important. One should ensure that there are minimal clashes at the interface.
6. Data-guided model construction for macromolecular complexes has gone well beyond knowledge-based ideas into the realms of molecular simulations with recent tools like MDFF, MELD, CryoFold leading the way. The authors should discuss about these advances and the complementarity of their methods to such integrative simulations in the discussions.

Reviewer #2:

Remarks to the Author:

In this work, Shor and Schneidman-Duhovny have presented an interesting extension of AF2 for modeling very large macromolecular assemblies. The manuscript is well-written and reflects considerable effort in their research endeavors. However, the following points are not adequately explored and therefore they should be carefully addressed in the next version of the paper.

1. The "integrative" aspect of the paper, involving data-driven modeling, warrants further exploration. A comprehensive discussion of the data-driven assembly modeling literature in the introduction would provide valuable context. Additionally, please elaborate on the incorporation of distance restraints throughout the paper, specifying if they serve as filters rather than drivers for the modeling. Explicitly presenting this distinction in the Introduction and Results sections, possibly including relevant information in Figure 1, would enhance clarity.

2. It would be really good to provide a visual illustration of the coupling between the PAE score and transformations, possibly including a complete procedure on a trimer as a supplementary figure.
3. For improved clarity, please include the run durations for the benchmark cases in a supplementary table.
4. In addition to the TM metric, please report the interface prediction qualities using either CASP- or CAPRI-based metrics.
5. Including the computed iPTM scores for the presented complexes would offer insights into how iPTM behaves on the subcomplex interfaces of large complexes.
6. I would appreciate clarification on the meaning of "predicted confidence" as mentioned in the manuscript.
7. When comparing to data-driven methods such as HADDOCK, ensure that the evaluation is performed with the inclusion of distance restraints to maintain fairness and accuracy.
8. For the CASP15 benchmark, please provide a comparison with the top performance generated by the community for the presented complexes to gauge whether the presented method performs better or worse than the community's best.
9. Please discuss the rationale behind using different subunit definitions for modeling different benchmarks and offer guidelines to users for selecting the appropriate method in real-case scenarios.
10. To strengthen the analysis, please include the distributions of all generated PAE scores for some representative runs and quantitatively evaluate whether the selected sub-complexes indeed exhibit significantly better PAE scores compared to eliminated binding modes.

Reviewer #3:

Remarks to the Author:

The authors present a strategy to predict the structure of large protein assemblies, leveraging on the ability of AlphaFold to produce accurate smaller complexes. It is shown that this strategy, embodied in the software CombFold, outperforms competing algorithms, and can be used in conjunction with distance restraints to improve predictions. In a proof-of-principle, it is also shown how CombFold appears to be usable to identify a subset of likely assembly stoichiometries, when experimental evidence is unavailable.

I find this work is interesting and results, supported by nice figures and appropriate references, very good. The abstract and main text are overall clearly written, though a few sentences feel slightly handwavy (see below for details). As the authors themselves indicate, hierarchical assembly using AlphaFold models has been already presented (MoLPC, published in Nature Communications in 2022), with result substantially beyond what at that time was state-of-the-art. The idea of incorporating distance restraints has also already explored AlphaLink. While the approach in MoLPC appears better than AlphaLink's, it is unclear to me to which extent the difference boils down to lack of feature implementation in AlphaLink, as opposed to a fundamental methodological superiority. Overall, in my view, while this is unquestionably a nice work, the improvements presented do not constitute the substantial advance I would expect from an article published in Nature Methods. My main comments are the following:

(1) Results reporting and comparison with MoLPC should be clarified. Specifically:

(1a) Considering that the authors state that their work is inspired by MoLPC, I think it is key to explicitly stress how their approach differs, what its crucial advantages are, and why. This will enable better gauging the level of novelty of this work.

(1b) In introduction, when discussing MoLPC, the authors state: "The recently developed MoLPC method relies on AlphaFold2 to produce configurations for pairs and triplets of chains and assemble them using Monte Carlo Tree Search. However, the approach is applicable mainly to homomeric complexes with a success rate of ~30%." Indeed, in the MoLPC article, a performance of 33% is reported. However, I understand this percentage referred to Top 1 high accuracy models (TM-score>0.8), not the acceptable quality ones (reported median score is 0.51).

(1c) In Table 1, I note that the performance of MoLC reported by the authors is 28% on Benchmark 3, i.e., the test set MoLPC was challenged with in its original publication. This success rate is less than what reported in the MoLPC article, what explains this inconsistency? As a side note, I also think the table caption should explicitly state what TM-score threshold has been used to produce the reported percentages.

(1d) Caption of Table 1 states "Slight modifications, such as dividing chains into domains, increased the success rate to 86%." This result reads as impressive as it is vague. I cannot find any data/method to substantiate this claim, that appears inconsistent with the text describing the results obtained on Benchmark 4. Could the authors clarify how this percentage was obtained?

(1e) As a minor point, I believe the original MoLPC article stated that problematic complexes are not the heteromeric ones, but those without a symmetry. While virtually all asymmetric complexes are also heteromeric, the opposite is not true. Could the authors investigate (or comment on) the effect of symmetry, as opposed to subunit composition?

(2) at p.2, it is stated: “Currently, common GPUs have no more than 20 gigabytes of memory, enabling the prediction of complexes up to 1,800 and 3,000 amino acids for AFMv2 and AFMv3, respectively. Also, as AFM memory usage increases roughly quadratically with the number of amino acids, any potential hardware advancements in the future are unlikely to have a significant impact.” While I can see the point the authors are making, I do not fully agree with it. This is because their observation only applies to situations where a single GPU is used, but modern ML applications exploit GPU’s ability of sharing memory. For instance, systems like the NVIDIA DGX A100 offer 320 GB memory and, about a months ago, NVIDIA has announced the DGX GH200, offering a staggering 144 TB memory. These architectures are not yet commonplace but, considering that pickup of new GPU technology has been rapid up to now so, it is reasonable to assume they will be within the next few years.

(3) When comparing their approach in applying experimental restraints with that presented in AlphaLink, the authors state: “This method requires subsampling of MSA to give more weight to distance restraints and is currently not applicable for complex structure prediction. The advantage of CombFold is that it can integrate additional information during the assembly stage”. Could the authors clarify whether the AlphaLink approach is inherently limited to monomers, or if it all boils down to an implementation detail? This will help understanding to which level the CombFold approach is fundamentally superior.

(4) at p.3., it is stated “The generation of candidate models is often performed by a Monte Carlo search of the conformational space.” I suggest this is substituted with “global optimization algorithms, as several existing protein docking software exploit other algorithms, e.g., genetic algorithms, or particle swarm optimization.

(5) In Methods, only average runtimes are reported. This sounds slightly anecdotal; I think it would be very useful for statistical information on runtimes to be provided, perhaps in Supplemental information (e.g., provide a histogram, or show how runtime scales with system size).

Author Rebuttal to Initial comments

We would like to thank the reviewers for their important comments, which have helped us to improve the manuscript significantly. Here and in the revised version we address all comments, with particular emphasis on the integrative aspect and comparison to additional methods - HADDOCK, AlphaLink, and RosettaFold2, to demonstrate the clear advance CombFold presents both with and without integrating crosslinks.

To demonstrate CombFold advance over end-to-end single step methods, and especially AFM, we have added a detailed figure and analysis (Fig. 7), illustrating how forcing AFM to predict

subcomplexes results in a higher number of correct pairwise interfaces that are not present in a single step AFM-predicted complete models of the complex. We have also added a detailed analysis in the Discussion to explain the novelty of CombFold over MoLPC.

Regarding the integrative aspect, several improvements were made in the integrative pipeline, including handling uncertainty and using scoring based on restraint satisfaction. We have added a benchmark that integrates crosslinks, based on Benchmark 2, and used it both to evaluate the effect of crosslinks integration on accuracy and for comparison of CombFold to other state-of-the-art methods that can integrate crosslinks - HADDOCK and AlphaLink. Results show that CombFold achieves substantially higher success rates when applied to large complexes compared to other methods and that crosslinks integration can result in a higher success rate with higher accuracy models (Fig. S2). We have also exposed the integrative aspect as part of the Colab Notebook so that crosslinks can be easily integrated by users.

Reviewer #1:

Remarks to the Author:

The article reports an AlphaFold-guided hierarchical docking procedure to construct models of protein assemblies, which also benefits from using experimental constraints. Beyond assembly construction, the tool is also used for stoichiometry prediction. The demonstration example on google collab is working. So my questions are primarily focused on the details of the tool.

1. Why is the tool better than AFM - what is making up for the failure of AFM and how ? Take one example and show it through all the steps to make this point.

There are four main reasons CombFold is expected to perform better than end-to-end (single step) AFM:

- (1) Using only a subset of subunits as input to AFM, "forces" it to find interfaces between those subunits, which would not happen when using the entire complex as input.*
- (2) Even if the pairwise interaction was not predicted correctly by AFM, it can still form indirectly during the assembly process.*
- (3) CombFold is applicable for large assemblies that can't be computed directly by AFM.*
- (4) CombFold can easily integrate distance restraints when available.*

As the first two reasons are non-trivial and related to the assembly algorithm, as suggested, we have added a figure (Fig. 7) that demonstrates on an example the advantage of CombFold and a paragraph explaining it in the Discussion:

"Two primary reasons account for CombFold's superior performance compared to AFM. First, the stage that generates pairwise subunit interactions enables us to find a higher number of

accurately predicted pairs. For example, for the early Pp module assembly intermediate of complex I, we find six pairwise interactions of Acceptable-quality (DockQ > 0.23, Fig. 7a). As a result in the assembly stage, several assembly pathways are possible because only four pairwise interactions that produce a spanning tree of all subunits are needed to assemble the complex. In contrast, AFM applied on the whole complex correctly predicts only three pairwise interactions (Fig. 7b). Second, even if the pairwise interaction was not predicted correctly by AFM, it can still form during the assembly process (Fig. 7d, subunits iii-v)."

2. For comparisons purposes what are the other strategies that can be utilized ? Beyond AFM, as an obvious choice comparison with this Rosetta tool (<https://pubmed.ncbi.nlm.nih.gov/33863889/>) should be made. The differences will have to be significant prove that the current work is not incremental.

We added a comparison to the newly released RosettaFold2 which supports multimers: "To further validate CombFold, we also used this benchmark for comparison to RosettaFold2(Baek et al. 2023). RosettaFold2 was not able to assemble most complexes (21/25), and among the assembled four complexes, only one had an acceptable quality model among the ten predicted structures, which translates to a success rate of 4% (Fig. S2)."

We have also benchmarked CombFold, HADDOCK, and AlphaLink using simulated crosslinks for Benchmark 2. The following text and Supp. figure were added to the Results:

"To further examine the contribution of crosslinking mass spectrometry data, we simulated crosslinks for Benchmark 2 (Methods) and compared the performance of CombFold with and without input crosslinks (Fig. S2). Integrating crosslinks increased the Top-1 success rate to 76% (compared to 64% without crosslinks). We compared CombFold to AlphaLink(Stahl et al. 2023) and HADDOCK(Dominguez et al. 2003) with the same set of crosslinks and obtained a success rate of 8% and 4%, respectively (Fig. S2)."

Therefore, compared to RosettaFold2, HADDOCK, and AlphaLink, we present substantial progress in our current work.

3. How's is uncertainty in the experimental data accounted for during the integration ? Simple harmonic potentials as constraints will have a number of overfitting issues if the data is not gaussian distributed (which is often the case) - how is this issue addressed ?

We agree with the comment and therefore implemented a routine to account for the uncertainty. We have added the following text to the Methods:

"CombFold accounts for the uncertainty in the crosslinking data and in the subunit structures as follows. The uncertainty in the data is accounted for by weighting each crosslink according to its

confidence based on the experimental evidence (w_1), such as the False Discovery Rate (FDR) (Leitner et al. 2020). To account for uncertainty in the subunit structures, each crosslink is weighted by the average AFM pLDDT score of the two crosslinked amino acids (w_2). The satisfaction ratio of a subcomplex is calculated as the sum of weights of satisfied distance restraints divided by the sum of weights of all restraints within the given subcomplex (Eq. 1)."

$$(1) \text{ satisfaction ratio} = \frac{\sum_{\text{satisfied}} w_1 * w_2}{\sum_{\text{all}} w_1 * w_2}$$

Regarding overfitting concerns, as AFM runs without crosslinks, all generated pairwise transformations are not based on crosslinks and therefore it is harder to overfit. Instead of harmonic potentials we use a hard distance threshold to determine if the crosslink is satisfied or not. However, we still changed how we consider distance restraints for subcomplexes - instead of a hard threshold on the percentage of satisfied crosslinks, we now integrate the crosslinks satisfaction ratio into the score. This score weighting should also prevent overfitting, and was proven effective on the new benchmark (Fig. S2). We have added the following text:

"The score of each subcomplex is multiplied by the satisfaction ratio. Consequently, as more restraints are fulfilled, the score increases, making it more probable for the subcomplex to avoid being filtered. A subcomplex is also filtered in the filtering stage if it violates some minimal percentage of its restraints (default 10%)"

4. Biophysical characterization (if available) of the assemblies with unknown structures will be even better. Mutational or biochemical validations are good nonetheless, but qualitative. Since no energy-based metric is optimized for model construction, these can potentially be used for validation. In the case experimental kD values are available for some complexes, the predicted models can be fed into the PRODIGY server and the statistic of the kD values from these CombFold models can be compared with those from the experiments.

We could not find a sufficient dataset of experimental kD values for meaningful testing. Therefore, we compared the kD values predicted by PRODIGY from the interfaces in experimental structures to the kD values predicted from the interfaces in our Top-1 models. We have added the following text and figure to the Results section:

"We examine whether the interface quality of CombFold models is sufficient for predicting dissociation constants (kD) between subunits. Because experimentally measured kD values are not available for the whole Benchmark, we compare the kD values predicted by PRODIGY (Xue et al. 2016) from the interfaces in experimental structures to the kD values predicted from the interfaces in the Top-1 model of CombFold. We find a strong correlation (Spearman $r=0.55$, Fig.

S4b), indicating that despite lower ICS scores, CombFold models are sufficiently accurate for estimating kD.”

5. Molprobitly scores telling about the quality of the produced models particularly at the interfaces will be important. One should ensure that there are minimal clashes at the interface.

Thank you for raising our awareness to this important issue. We have added to CombFold a relaxation step that is also used in AlphaFold2. The following paragraph and a figure were added to the Methods section:

“Relaxation. As a result of using representative subunit structures, CombFold may produce structures with steric clashes in interfaces, mainly in side-chains. Therefore, it is recommended to perform an extra step of relaxation of the structure by gradient descent using the Amber(Hornak et al. 2006) force field similar to AlphaFold. This step substantially reduces the clashscore calculated by Molprobitly(Williams et al. 2018) (Fig. S4c) while not affecting the structure significantly (change in C α RMSD <1Å in all targets of Benchmark 2).”

6. Data-guided model construction for macromolecular complexes has gone well beyond knowledge-based ideas into the realms of molecular simulations with recent tools like MDFF, MELD, CryoFold leading the way. The authors should discuss about these advances and the complementarity of their methods to such integrative simulations in the discussions.

We completely agree with the reviewer and added a discussion of data-driven simulation approaches that use Bayesian approach to data interpretation:

“The Bayesian approach that can account for most sources of uncertainty in data without over-fitting is often used for determining structural ensembles(Rieping et al. 2005). This approach estimates the probability of a model, given information available about the system, including both prior knowledge and newly acquired experimental data. It was successfully integrated into data-driven MD simulations and adopted for multiple types of data, including cryo-EM density maps(Bonomi et al. 2018) and contact or distance information from multiple sources(MacCallum et al. 2015). Our current implementation can integrate distance-based information into the assembly stage and generate multiple models that are consistent with the data. Moreover, models generated by CombFold can be used as starting points for generating dynamic ensembles using data-driven simulation approaches, such as CryoFold(Shekhar et al. 2021; Chang et al. 2023).”

Reviewer #2:

Remarks to the Author:

In this work, Shor and Schneidman-Duhovny have presented an interesting extension of AF2 for modeling very large macromolecular assemblies. The manuscript is well-written and reflects considerable effort in their research endeavors. However, the following points are not adequately explored and therefore they should be carefully addressed in the next version of the paper.

1. The "integrative" aspect of the paper, involving data-driven modeling, warrants further exploration. A comprehensive discussion of the data-driven assembly modeling literature in the introduction would provide valuable context. Additionally, please elaborate on the incorporation of distance restraints throughout the paper, specifying if they serve as filters rather than drivers for the modeling. Explicitly presenting this distinction in the Introduction and Results sections, possibly including relevant information in Figure 1, would enhance clarity.

We have further extended the integrative aspect of the paper as follows.

1. A paragraph with relevant references was added to the Introduction section:

“The integrative modeling workflow iterates through four stages that convert input information into an output model: (1) gathering data; (2) scoring (representing and translating the data into spatial restraints); (3) sampling; and (4) validating the model (Alber et al. 2007; Russel et al. 2012). The sampling of candidate models is often performed by global data driven optimization algorithms, such as Monte Carlo or genetic algorithms. The input information contributes to a scoring function, either for ranking or filtering generated structural models or for directly guiding the sampling process. Integrative structure modeling is applicable to large and heterogeneous systems (Rout and Sali 2019), such as the ~52 MDa Nuclear Pore Complex (Kim et al. 2018).”

2. Support of data and structure uncertainty and improved scoring along with additional explanations were added to the Methods section (see also response to Reviewer #1, comment 3):

“CombFold accounts for the uncertainty in the crosslinking data and in the subunit structures as follows. The uncertainty in the data is accounted for by weighting each crosslink according to its confidence based on the experimental evidence (w_1), such as the False Discovery Rate (FDR) (Leitner et al. 2020). To account for uncertainty in the subunit structures, each crosslink is weighted by the average AFM pLDDT score of the two crosslinked amino acids (w_2). The satisfaction ratio of a subcomplex is calculated as the sum of weights of satisfied distance restraints divided by the sum of weights of all restraints within the given subcomplex (Eq. 1). The score of each subcomplex is multiplied by the satisfaction ratio. Consequently, as more

restraints are fulfilled, the score increases, making it more probable for the subcomplex to avoid being filtered. A subcomplex is also filtered in the filtering stage if it violates some minimal percentage of its restraints (default 10%)."

3. A new validation of the effect of incorporating crosslinking data on Benchmark 2 complexes was added (Results & Fig. S2), including comparison to AlphaLink and HADDOCK (as detailed in comment 7 below).

"To further examine the contribution of crosslinking mass spectrometry data, we simulated crosslinks for Benchmark 2 (Methods) and compared the performance of CombFold with and without input crosslinks (Fig. S2). Integrating crosslinks increased the Top-1 success rate to 76% (compared to 64% without crosslinks)."

4. As suggested, we have also added a figure extending Fig. 1, describing the filtering process and the incorporation of distance restraints in it (Fig. S8).

2. It would be really good to provide a visual illustration of the coupling between the PAE score and transformations, possibly including a complete procedure on a trimer as a supplementary figure.

We have added a detailed example of the assembly graph and assembly order along with PAE and DockQ scores for a complex with five subunits in a new Fig. 7. The figure also illustrates the advantage of hierarchical assembly. The following text was added to the Discussion:

"The higher accuracy of CombFold compared to AFM is attributed to two main factors. First, the stage that generates pairwise subunit interactions enables us to find a higher number of accurately predicted pairs. For example, for the early Pp module assembly intermediate of complex I, we find six pairwise interactions of Acceptable-quality (DockQ > 0.23, Fig. 7a). As a result in the assembly stage, several assembly pathways are possible because only four interactions that produce a spanning tree of all subunits are needed to assemble the complex. In contrast, AFM applied on the whole complex correctly predicts only three pairwise interactions (Fig. 7b). Second, even if the pairwise interaction was not predicted correctly by AFM, it can still form during the assembly process (Fig. 7d, subunits iii-v)."

3. For improved clarity, please include the run durations for the benchmark cases in a supplementary table.

We have added a detailed Supplementary Table S4 of run durations, as well as a figure showing the number of complex subunits vs. time (Fig S7).

4. In addition to the TM metric, please report the interface prediction qualities using either CASP- or CAPRI-based metrics.

We have used an Interface Contact Similarity (ICS) measure that is also used by CASP/CAPRI assessors. The following text and figure were added to the Results section:

“While TM-score is a measure of global accuracy, to assess the accuracy of subunit interfaces, we calculate the Interface Contact Similarity (ICS) score (Lafita et al. 2018) that is also used in the CASP/CAPRI complex assessment. Similarly to the TM-score, ICS values are in the range of [0-1], however the ICS scores are usually lower compared to TM-scores, indicating that a model with high global accuracy may still have low-quality interfaces and contacts. We find that CombFold Top-1 models have variable ICS scores (Fig. S4a). Moreover, AFM models have higher scores compared to CombFold. The lower ICS scores of CombFold can be attributed to usage of representative subunit structures instead of the ones produced by pairwise AFM. In addition, some of the interfaces in the CombFold models are not a result of pairwise AFM prediction, but a by-product of the assembly process, and therefore have lower quality.”

5. Including the computed ipTM scores for the presented complexes would offer insights into how ipTM behaves on the subcomplex interfaces of large complexes.

We have added a Supplementary figure with several pairwise scoring functions, ipTM among them, and their correlation to RMSD of the pairwise interaction model:

“Multiple possibilities for scoring were considered, including PAE, the minimal PAE, the interface PAE of the interacting amino acids only, ipTM, and ipLDDT which is widely used (Yin et al. 2022; Bryant et al. 2022; He et al. 2023). All scores had a comparable correlation with Ca RMSD (Pearson r of $\sim 0.5-0.6$, Fig. S6). The advantage of our PAE-based score is that incorrect interfaces consistently have low scores (Fig. S6e).”

6. I would appreciate clarification on the meaning of "predicted confidence" as mentioned in the manuscript.

We have added a paragraph to the Methods:

“Predicted confidence. CombFold predicts the confidence of the assembled structure as a weighted score of the pairwise transformation scores (S_T) used in the assembly stage. To calculate the weight of a given transformation (W_T), we split the complex into two subcomplexes using the transformation and the complex assembly tree. The weight of the transformation is the number of amino acids in the smaller subcomplex. The idea is that some transformations have a larger effect on the final global structure of the complex, as they affect a larger number of amino

acids. The final score is normalized by the total weight of all the transformations used in the assembly stage (Eq. 2)."

$$(2) \quad \text{predicted confidence} = \frac{\sum_T W_T * S_T}{\sum_T W_T}$$

7. When comparing to data-driven methods such as HADDOCK, ensure that the evaluation is performed with the inclusion of distance restraints to maintain fairness and accuracy.

We have benchmarked CombFold, HADDOCK, and AlphaLink using simulated crosslinks for Benchmark 2. The following text and Supp. figure were added to the Results:

"To further examine the contribution of crosslinking mass spectrometry data, we simulated crosslinks for Benchmark 2 (Methods) and compared the performance of CombFold with and without input crosslinks (Fig. S2). Integrating crosslinks increased the Top-1 success rate to 76% (compared to 64% without crosslinks). We compared CombFold to AlphaLink(Stahl et al. 2023) and HADDOCK(Dominguez et al. 2003) with the same set of crosslinks and obtained a success rate of 8% and 4%, respectively (Fig. S2)."

8. For the CASP15 benchmark, please provide a comparison with the top performance generated by the community for the presented complexes to gauge whether the presented method performs better or worse than the community's best.

We have compared CombFold to AFM, MoLPC and the CASP15 top performing groups and servers (Table S3). The following text was added:

"AFMv2 was able to produce acceptable quality models only for one of these targets. MoLPC was successful only in three targets. CombFold performance is comparable with the top-ranked groups and servers that participated in CASP15 competition (Table S3). For almost all targets, different methods were able to produce High-quality models (TM-score > 0.85). There was a notable exception in the case of complex T1115, where CombFold managed to achieve a higher TM-score compared to the top-ranked server's performance (0.72 vs. 0.66). However, human predictors submitted models with higher TM-score."

9. Please discuss the rationale behind using different subunit definitions for modeling different benchmarks and offer guidelines to users for selecting the appropriate method in real-case scenarios.

Extended discussion and explanations in Methods.

Regarding different subunit definitions:

“In Benchmark 4, each subunit was defined as a single chain according to definitions supplied by CASP. The two targets which are long single chains (T1165, T1169) were divided into subunits according to IUPred3(Erdős et al. 2021). The predicted disordered regions connecting the domains were not included in the prediction. In all other benchmarks a full chain was used as a subunit as defined in the SEQRES segment of the PDB entry for almost all cases. Due to a high number of long chains in Benchmarks 2 and 3, we opted for a simple split procedure without relying on predicted disorder regions.”

Regarding selecting the appropriate method in real-case scenarios:

“In case a chain is too long for modeling with other chains, or if it is known to contain a linker, it is best to divide it into structural domains based on predicted disordered regions using predictors, such as IUPred3(Erdős et al. 2021).”

10. To strengthen the analysis, please include the distributions of all generated PAE scores for some representative runs and quantitatively evaluate whether the selected sub-complexes indeed exhibit significantly better PAE scores compared to eliminated binding modes.

Thank you for this suggestion. We have added a figure with PAE distributions for all pairs vs. the pairs that were used in Top-1 CombFold models (Fig. S6f) and the following text to Methods:

“Our analysis of average PAE distributions of all AFM pairwise interaction modes vs. the ones that were selected for Top-1 assembly models revealed that CombFold indeed selects the interactions with lower PAE scores (Fig. S6f).”

Reviewer #3:

Remarks to the Author:

The authors present a strategy to predict the structure of large protein assemblies, leveraging on the ability of AlphaFold to produce accurate smaller complexes. It is shown that this strategy, embodied in the software CombFold, outperforms competing algorithms, and can be used in conjunction with distance restraints to improve predictions. In a proof-of-principle, it is also shown how CombFold appears to be usable to identify a subset of likely assembly stoichiometries, when experimental evidence is unavailable.

I find this work interesting and results, supported by nice figures and appropriate references, very good. The abstract and main text are overall clearly written, though a few sentences feel slightly handwavy (see below for details). As the authors themselves indicate, hierarchical assembly using AlphaFold models has been already presented (MoLPC, published in Nature Communications in 2022), with results substantially beyond what at that time was state-of-the-art. The idea of incorporating distance restraints has also already been explored in AlphaLink. While the approach in MoLPC appears better than AlphaLink's, it is unclear to me to which extent the difference boils down to lack of feature implementation in AlphaLink, as opposed to a fundamental methodological superiority. Overall, in my view, while this is unquestionably a nice work, the improvements presented do not constitute the substantial advance I would expect from an article published in Nature Methods. My main comments are the following:

(1) Results reporting and comparison with MoLPC should be clarified. Specifically:

(1a) Considering that the authors state that their work is inspired by MoLPC, I think it is key to explicitly stress how their approach differs, what its crucial advantages are, and why. This will enable better gauging the level of novelty of this work.

We have added a paragraph to the discussion that explores this:

"The superior performance of CombFold compared to MoLPC can be attributed to several factors. First, by employing a more exhaustive combinatorial assembly algorithm, and implementing clustering during assembly, we are able to better enumerate the many possible interactions between subunits, resulting in a higher number of accurate assemblies. Second, the enumeration process of CombFold is more strongly based on the confidence score of each transformation, which correlates to accuracy (Fig. S6), and therefore, CombFold is able to select the more confident and accurately predicted interactions (Fig. S6f). Third, the usage of a unified representation results in each subunit model being the most confident AFM-generated model of this subunit, which results in an overall more accurate complex structure. Lastly, implementation details such as a more relaxed steric clashes filtering stage, and AFM prediction for groups of

more than three subunits efficiently, can be more effective when implementing assembly-based methods.”

(1b) In introduction, when discussing MoLPC, the authors state: “The recently developed MoLPC method relies on AlphaFold2 to produce configurations for pairs and triplets of chains and assemble them using Monte Carlo Tree Search. However, the approach is applicable mainly to homomeric complexes with a success rate of ~30%.”. Indeed, in the MoLPC article, a performance of 33% is reported. However, I understand this percentage referred to Top 1 high accuracy models (TM-score>0.8), not the acceptable quality ones (reported median score is 0.51).

(1c) In Table 1, I note that the performance of MoLPC reported by the authors is 28% on Benchmark 3, i.e., the test set MoLPC was challenged with in its original publication. This success rate is less than what reported in the MoLPC article, what explains this inconsistency? As a side note, I also think the table caption should explicitly state what TM-score threshold has been used to produce the reported percentages.

We have added a paragraph to the Methods explaining these inconsistencies when comparing to MoLPC:

“MoLPC evaluation used a TM-score above 0.8 to define a High-quality prediction. Here, we use the same definition of High-quality prediction. We find that a prediction with a TM-score of 0.7 can have a correct global shape (Fig. 3h, 4c,d, 7b). Therefore, we define an additional Acceptable-quality category for predictions with a TM-score above 0.7. In the original MoLPC publication, the success rate was calculated as a fraction of benchmark cases with a High-quality prediction out of cases where at least one assembly was obtained. Note that MoLPC was able to obtain some predictions for 91 out of 175 Benchmark 3 cases. Here, we define a success rate as a fraction of benchmark cases with an Acceptable-quality prediction out of all benchmark cases. In addition, while MoLPC has presented separate success rates for AFM-based or FoldDock-based pipelines, we have considered results from both pipelines in our calculated success rate. We recalculated the success rate of MoLPC according to our definitions, resulting in slightly different values.”

We also added a clarification in Table 1 caption. “The success rate is defined as the fraction of benchmark cases with a model with a TM-score above 0.7 among the Top-N best-scoring predictions.”

(1d) Caption of Table 1 states “Slight modifications, such as dividing chains into domains, increased the success rate to 86%.”. This result reads as impressive as it is vague. I cannot find any data/method to substantiate this claim, that appears inconsistent with the text describing the results obtained on Benchmark 4. Could the authors clarify how this percentage was obtained?

A clarification was added in the Results section:

“The automatic pipeline was able to generate Acceptable-quality models for four of the seven targets (Fig. 4a-d, T1192, H111, T1115, T1165), which translates to a success rate of 57%. For the remaining three targets, using manual division of the chains into subunits as described below, enabled to generate a High-quality model for two additional targets (Fig. 4f-g, T1169, H1137), increasing the success rate to 86%.“

(1e) As a minor point, I believe the original MoLPC article stated that problematic complexes are not the heteromeric ones, but those without a symmetry. While virtually all asymmetric complexes are also heteromeric, the opposite is not true. Could the authors investigate (or comment on) the effect of symmetry, as opposed to subunit composition?

We completely agree and have added a comment on our performance on asymmetric complexes:

“While heteromeric complexes are asymmetric by definition, they can include local symmetry resulting from multiple copies of one or more subunits(Duarte et al. 2022). Benchmark 3 contains four fully asymmetric complexes (without multiple subunit copies) and CombFold was able to assemble three with Acceptable-quality. The performance of CombFold on asymmetric structures is assessed on Benchmarks 1 & 2, which are almost entirely asymmetric (Fig S1).“

(2) at p.2, it is stated: “Currently, common GPUs have no more than 20 gigabytes of memory, enabling the prediction of complexes up to 1,800 and 3,000 amino acids for AFMv2 and AFMv3, respectively. Also, as AFM memory usage increases roughly quadratically with the number of amino acids, any potential hardware advancements in the future are unlikely to have a significant impact.” While I can see the point the authors are making, I do not fully agree with it. This is because their observation only applies to situations where a single GPU is used, but modern ML applications exploit GPU’s ability of sharing memory. For instance, systems like the NVIDIA DGX A100 offer 320 GB memory and, about a months ago, NVIDIA has announced the DGX GH200, offering a staggering 144 TB memory. These architectures are not yet commonplace but, considering that pickup of new GPU technology has been rapid up to now so, it is reasonable to assume they will be within the next few years.

We have modified the sentences as follows:

“Also, as AFM memory usage increases roughly quadratically with the number of amino acids, potential hardware advancements in the future need to be substantial to resolve these issues. It is also worth noting that hardware advancements take time to be implemented and become accessible. This limits the practical capability of many researchers to predict structures of large size, leaving many macromolecular complexes without a structure prediction.”

(3) When comparing their approach in applying experimental restraints with that presented in AlphaLink, the authors state: “This method requires subsampling of MSA to give more weight to distance restraints and is currently not applicable for complex structure prediction. The advantage of CombFold is that it can integrate additional information during the assembly stage”. Could the authors clarify whether the AlphaLink approach is inherently limited to monomers, or if it all boils down to an implementation detail? This will help understanding to which level the CombFold approach is fundamentally superior.

After our initial submission, a preprint was released that applies AlphaLink for complexes. We have created a new benchmark based on Benchmark 2 with simulated crosslinks, to compare CombFold to crosslinks-based structure prediction methods, including AlphaLink. The results show that AlphaLink is not yet applicable to large complexes (Fig. S2c).

We have also changed the referenced text.

“...and is currently applicable for complexes with less than 3,000 amino acids”

In the revised manuscript we have also added a new Figure 7 and analysis to demonstrate CombFold advantage over AFM when computing large complexes. The analyzed advantages also apply to other structure prediction methods such as AlphaLink.

“The higher accuracy of CombFold compared to AFM is attributed to two main factors. First, the stage that generates pairwise subunit interactions enables us to find a higher number of accurately predicted pairs. For example, for the early Pp module assembly intermediate of complex I, we find six pairwise interactions of Acceptable-quality (DockQ > 0.23, Fig. 7a). As a result in the assembly stage, several assembly pathways are possible because only four interactions that produce a spanning tree of all subunits are needed to assemble the complex. In contrast, AFM applied on the whole complex correctly predicts only three pairwise interactions (Fig. 7b). Second, even if the pairwise interaction was not predicted correctly by AFM, it can still form during the assembly process (Fig. 7d, subunits iii-v). This also applies to other end-to-end (single step) methods, such as RosettaFold2 and AlphaLink.”

(4) at p.3., it is stated “The generation of candidate models is often performed by a Monte Carlo search of the conformational space.” I suggest this is substituted with “global optimization algorithms, as several existing protein docking software exploit other algorithms, e.g., genetic algorithms, or particle swarm optimization.

We have changed this sentence to:

“The sampling of candidate models is often performed by global data driven optimization algorithms, such as Monte Carlo or genetic algorithms.”

(5) In Methods, only average runtimes are reported. This sounds slightly anecdotal; I think it would be very useful for statistical information on runtimes to be provided, perhaps in Supplemental information (e.g., provide a histogram, or show how runtime scales with system size).

We have added a detailed Supplementary Table S4 of run durations, and as suggested, a figure showing the number of complex subunits vs. time (Fig S7).

Decision Letter, first revision:

Dear Dina,

Thank you for submitting your revised manuscript "Predicting structures of large protein assemblies using combinatorial assembly algorithm and AlphaFold2" (N METH-A52611A). It has now been seen by the original referees and their comments are below. The reviewers find that the paper has improved in revision, and therefore we'll be happy in principle to publish it in Nature Methods, pending minor revisions to satisfy the referees' final requests and to comply with our editorial and formatting guidelines.

TRANSPARENT PEER REVIEW

Nature Methods offers a transparent peer review option for new original research manuscripts submitted from 17th February 2021. We encourage increased transparency in peer review by publishing the reviewer comments, author rebuttal letters and editorial decision letters if the authors agree. Such peer review material is made available as a supplementary peer review file. Please state in the cover letter 'I wish to participate in transparent peer review' if you want to opt in, or 'I do not wish to participate in transparent peer review' if you don't. Failure to state your preference will result in delays in accepting your manuscript for publication.

ORCID

Sincerely,
Arunima

Arunima Singh, Ph.D.
Senior Editor
Nature Methods

Reviewer #1 (Remarks to the Author):

My points are resolved, and I am happy to support the article.

Reviewer #2 (Remarks to the Author):

I would like to thank the authors for addressing all my concerns. The paper is in a shape to be accepted for a publication.

Reviewer #3 (Remarks to the Author):

The authors have addressed my comments, and the text has improved as a result. In particular, the text better clarifies how the modelling method presented here differs from the state-of-the-art. While the idea of producing models of large protein assemblies from subcomplexes produced by AlphaFold is not novel, the algorithmic improvements presented go beyond incremental modifications of competing

approaches. Overall, results presented make a compelling case for an effective way to address this modelling challenge.

I only have few remaining comments.

1) As I detailed in my first review, I think that the point on large memory requirements hindering the application of AlphaFold to large complexes is not very strong. It seems that the authors have opted to extend the text with a statement directly countering my observations. While I maintain that I expect memory requirements not to be as much of a problem as the authors are stating, I am happy to be proven wrong. To solve this disagreement, perhaps the authors could estimate the memory required for the modelling of the complexes they are testing, and compare that with memory offered by existing GPU technology with, e.g., the 320 GB offered by the commercially available NVIDIA DGX A100.

2) The updated caption of Table 1 reads “[...] slight modifications, such as dividing chains into domains, increased the success rate to 86%.” Using “slight” and “such as” still makes the caption read handwavy. I suggest something to the effect of “Manual subdivision of proteins into domains led to an increased success rate of 86%”. If other modifications were required, state them explicitly.

3) A CA-CA cutoff distance of 25Å was adopted to simulate lysine cross-links. I suggest adding a reference to justify why this distance was chosen. I assume it approximates experiments with BS3? While I think that the approach chosen is suitable, I note that this criterion already comes with its fair share of false positives and negatives (does not explicitly accommodate side-chain flexibility and cross-linker physical nature).

Author Rebuttal, first revision:

1) As I detailed in my first review, I think that the point on large memory requirements hindering the application of AlphaFold to large complexes is not very strong. It seems that the authors have opted to extend the text with a statement directly countering my observations. While I maintain that I expect memory requirements not to be as much of a problem as the authors are stating, I am happy to be proven wrong. To solve this disagreement, perhaps the authors could estimate the memory required for the modelling of the complexes they are testing, and compare that with memory offered by existing GPU technology with, e.g., the 320 GB offered by the commercially available NVIDIA DGX A100.

We agree with the reviewer that the memory issue might be resolved in newer GPU cards and we have updated the text as follows:

“We estimate that in a few years, GPU cards with sufficient memory will become widely available. However, as AFM memory usage increases roughly quadratically with the number of amino acids⁷, this

currently limits the practical capability of many researchers to predict structures of large size, leaving many macromolecular complexes without a structure prediction.”

2) The updated caption of Table 1 reads “[...] slight modifications, such as dividing chains into domains, increased the success rate to 86%.” Using “slight” and “such as” still makes the caption read handwavy. I suggest something to the effect of “Manual subdivision of proteins into domains led to an increased success rate of 86%”. If other modifications were required, state them explicitly.

We have updated the caption as suggested to: “Manual subdivision of proteins into domains led to an increased success rate of 86%”

3) A CA-CA cutoff distance of 25Å was adopted to simulate lysine cross-links. I suggest adding a reference to justify why this distance was chosen. I assume it approximates experiments with BS3? While I think that the approach chosen is suitable, I note that this criterion already comes with a its fair share of false positives and negatives (does not explicitly accommodate side-chain flexibility and cross-linker physical nature).

We have added these two references that measure cross-linked distances in crystal structures:

1. The beginning of a beautiful friendship: cross-linking/mass spectrometry and modelling of proteins and multi-protein complexes
2. Chemical cross-linking/mass spectrometry targeting acidic residues in proteins and protein complexes

Note that due to the manuscript size limitations, this part is now in a Supplementary Note 2.

Final Decision Letter:

Dear Dina,

I am pleased to inform you that your Article, "CombFold: Predicting structures of large protein assemblies using combinatorial assembly algorithm and AlphaFold2", has now been accepted for publication in Nature Methods. The received and accepted dates will be May 17, 2023 and January 9, 2024. This note is intended to let you know what to expect from us over the next month or so, and to let you know where to address any further questions.

Over the next few weeks, your paper will be copyedited to ensure that it conforms to Nature Methods style. Once your paper is typeset, you will receive an email with a link to choose the appropriate publishing options for your paper and our Author Services team will be in touch regarding any additional

information that may be required. It is extremely important that you let us know now whether you will be difficult to contact over the next month. If this is the case, we ask that you send us the contact information (email, phone and fax) of someone who will be able to check the proofs and deal with any last-minute problems.

Please note that *Nature Methods* is a Transformative Journal (TJ). Authors may publish their research with us through the traditional subscription access route or make their paper immediately open access through payment of an article-processing charge (APC). Authors will not be required to make a final decision about access to their article until it has been accepted. [Find out more about Transformative Journals](https://www.springernature.com/gp/open-research/transformative-journals)

You may wish to make your media relations office aware of your accepted publication, in case they consider it appropriate to organize some internal or external publicity. Once your paper has been scheduled you will receive an email confirming the publication details. This is normally 3-4 working days in advance of publication. If you need additional notice of the date and time of publication, please let the production team know when you receive the proof of your article to ensure there is sufficient time

to coordinate. Further information on our embargo policies can be found here:
<https://www.nature.com/authors/policies/embargo.html>

If you are active on Twitter/X, please e-mail me your and your coauthors' handles so that we may tag you when the paper is published.

Best regards,
Arunima

Arunima Singh, Ph.D.
Senior Editor
Nature Methods